# The phased pan-genome of tetraploid European potato

Hequan Sun[1,2,3,4,10], Sergio Tusso[3,4,10], Craig I. Dent[4,5], Manish Goel[3,4], Raúl Y. Wijfjes[3], Lisa C. Baus[3], Xiao Dong[4], José A. Campoy[4,9], Ana Kurdadze[3], Birgit Walkemeier[4], Christine Sänger[4], Bruno Huettel[6], Ronald C. B. Hutten[7], Herman J. van Eck[7], Klaus J. Dehmer[5,8] & Korbinian Schneeberger[3,4,5] ✉

Potatoes were first brought to Europe in the sixteenth century[1,2]. Two hundred years later, one of the species had become one of the most important food sources across the entire continent and, later, even the entire world[3]. However, its highly heterozygous, autotetraploid genome has complicated its improvement since then[4–7]. Here we present the pan-genome of European potatoes generated from phased genome assemblies of ten historical potato cultivars, which includes approximately 85% of all haplotypes segregating in Europe. Sequence diversity between the haplotypes was extremely high (for example, 20× higher than in humans), owing to numerous introgressions from wild potato species. By contrast, haplotype diversity was very low, in agreement with the population bottlenecks caused by domestication and transition to Europe. To illustrate a practical application of the pan-genome, we converted it into a haplotype graph and used it to generate phased, megabase-scale pseudo-genome assemblies of commercial potatoes (including the famous French fries potato 'Russet Burbank') using cost-efficient short reads only. In summary, we present a nearly complete pan-genome of autotetraploid European potato, we describe extraordinarily high sequence diversity in a domesticated crop, and we outline how this resource might be used to accelerate genomics-assisted breeding and research.

Cultivated potato (*Solanum tuberosum* subspecies (ssp.) *tuberosum*) is the most important non-cereal food crop, feeding over a billion people worldwide[8]. But despite this importance, potato improvement has not been as successful compared with other species. The main reason for this is its autotetraploid genome, which makes conventional, cross-based breeding difficult and poses challenges for any type of genome assembly and analysis. So far, only three complete and haplotype-resolved genome assemblies of tetraploid cultivars have been generated[5–7]. Although long DNA sequencing reads were sufficient to separate and assemble the sequences of different haplotypes (that is, individual chromosome molecules), regions that were shared between the haplotypes (partly homozygous regions) could not be resolved with long reads alone[4]. To achieve this, all three assemblies required tedious generation of recombinant offspring populations in addition to sequencing the genomes. Afterwards, the three assemblies revealed exceptionally high genetic diversity (about 1 difference in 50 base pairs (bp)), which is around 20 times higher than in humans and four times higher than in wild *Arabidopsis thaliana* plants[9,10], highlighting the complexity of tetraploid potato genomes.

Many different potato species have been domesticated in the Andean highlands in South America for about 10,000 years and were first shipped to Europe around 1560 (refs. 1,2,11) (Fig. 1a). But only a single species successfully adapted to Europe's seasonal temperatures and day lengths[3,12]. By the end of the eighteenth century, this species (the European potato) had become the main staple crop in many parts of the continent. However, in the middle of nineteenth century, susceptibility to *Phytophthora infestans* (a fungus-like microorganism that causes the potato late blight disease in which infected tubers rot in the ground) led to devastating famines in Ireland and other countries[13], and marked the onset of modern potato breeding in Europe. The first breeding programmes used locally grown varieties, relying primarily on crossing existing cultivars. Later efforts in the twentieth century saw further introgressions of foreign haplotypes, mainly to introduce new resistance alleles[6,7,14,15]. The history of European potatoes—domestication in South America, introduction to Europe, adaptation to new environments, germplasm loss during epidemics and inbreeding during the past approximately 150 years—suggests a series of severe genetic bottlenecks that reduced genomic diversity, similar to other crops[16]. Although the severe bottlenecks and extremely high genetic

[1]MOE Key Laboratory for Intelligent Networks & Network Security, Faculty of Electronic and Information Engineering, Xi'an Jiaotong University, Xi'an, China. [2]School of Automation Science and Engineering, Faculty of Electronic and Information Engineering, Xi'an Jiaotong University, Xi'an, China. [3]Faculty of Biology, LMU Munich, Planegg-Martinsried, Germany. [4]Department of Chromosome Biology, Max Planck Institute for Plant Breeding Research, Cologne, Germany. [5]CEPLAS: Cluster of Excellence on Plant Sciences, Heinrich-Heine-University, Düsseldorf, Germany. [6]Max Planck Genome Center, Max Planck Institute for Plant Breeding Research, Cologne, Germany. [7]Plant Breeding, Wageningen University & Research, Wageningen, The Netherlands. [8]Leibniz Institute of Plant Genetics and Crop Plant Research (IPK), Gross Luesewitz, Germany. [9]Present address: Department of Agronomical Engineering, Institute of Plant Biotechnology, Universidad Politécnica de Cartagena, Cartagena, Spain. [10]These authors contributed equally: Hequan Sun, Sergio Tusso. ✉e-mail: k.schneeberger@lmu.de

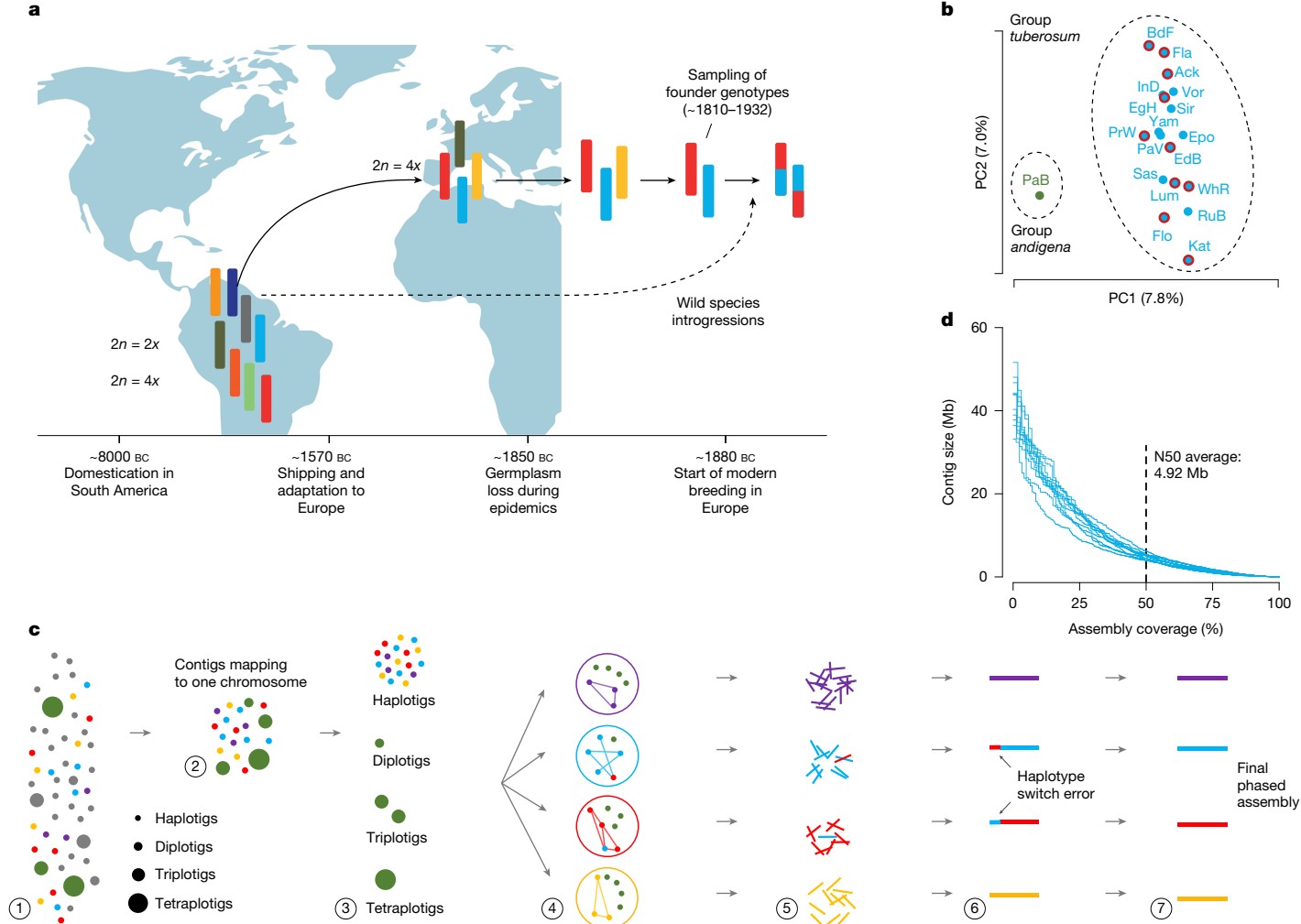

**Fig. 1 | History, genomic variation and phased genome assembly of European potato. a**, Potatoes were domesticated around 10,000 years ago in South America, and were introduced to Europe for the first time around 1570. After potato became a major crop across Europe, in the middle of the nineteenth century, lack of resistance against *P. infestans* led to the Irish Famine and the loss of susceptible genotypes. Modern potato breeding started in Europe around 1880. Since then, potato breeding has been mostly based on inter-crossing European potatoes with the exception of a few wild introgressions. **b**, Selection of ten cultivars for pan-genome construction, after principal component analysis of the genetic variation among nineteen cultivars. The first two principal components (PC1 and PC2) capture the major axes of genetic variation. 'Papa bonita' was identified as an outlier that probably belongs to a different subspecies, *S. tuberosum* ssp. *andigena*[20], and it was not included in the subsequent breeding programmes in Europe[3]. **c**, Genome assembly workflow.

(1) Initial whole-genome contig-level assembly is generated with long reads. (2) Contigs are sorted to chromosomes using homology. (3) Short-read alignments define haplotigs, diplotigs, triplotigs and tetraplotigs. (4) For each chromosome, haplotigs are phased into four haplo-groups with Hi-C data. Diplotigs, triplotigs and tetraploid are assigned to two, three or four of the haplo-groups using Hi-C data. (5) Long reads are re-aligned to the contigs of each haplo-group. (6) Long reads of each haplo-group are re-assembled and contigs are scaffolded to chromosome level using Hi-C data. (7) Hi-C contact maps are used to correct haplotype switch errors. **d**, Contig contiguities of the genome assemblies of the ten potato cultivars (plus 'Russet Burbank'), showing contig N50 from 4.0 to 6.2 Mb. Panel **a** was created using BioRender (https://biorender.com/v97e168). Ack, 'Ackersegen'; BdF, 'Belle de Fontenay'; EdB, 'Edzell Blue'; EgH, 'Eigenheimer'; Fla, 'Flava'; Flo, 'Flourball'; Kat, 'Katahdin'; Lum, 'Lumper'; PrW, 'Prof. Wohltmann'; WhR, 'White Rose'.

diversity observed in European potato are not necessarily a contradiction, this raised questions about the actual impact of these bottlenecks[14].

To accelerate genomics-assisted breeding and research in potato, we assembled a haplotype-resolved pan-genome of European potato. Using a specialized assembly pipeline for autotetraploid genomes, we assembled the genomes of ten historical cultivars. Despite high sequence diversity, haplotype diversity was strikingly limited, with an average of only nine out of 40 potentially different haplotypes per genomic region. This combination of a highly diverse, but restricted, haplotype space can be explained by the series of strong genetic bottlenecks combined with admixture with wild potato species that led to extensive sequence diversity. Pan-genome analysis suggested that the assembly already captures around 85% of the total diversity

in European potato. By converting the pan-genome into a genome graph and using it as a reference for short-read alignments, we could reconstruct megabase-scale haplotype blocks of modern cultivars using short-read sequencing only, demonstrating the utility of this pan-genome for genomics-assisted breeding and research in European potatoes.

## Phased pan-genome of European potato

To characterize the genetic diversity present at the onset of European potato breeding, we searched the Wageningen University Potato Pedigree Database (including pedigree data from over 9,500 potato samples as of July 2021) for cultivars grown in the nineteenth century or those foundational to modern breeding efforts[17,18] (Supplementary Table 1).

Of 164 cultivars that fitted our criteria, 19 were available in the Gross Lüsewitz Potato Collections (GLKS) of the Leibniz Institute of Plant Genetics and Crop Plant Research Gene Bank, where such material is being preserved long term[19] (Supplementary Table 2). Whole-genome short-read sequencing of the 19 samples revealed that one sample was from a different subspecies, *S. tuberosum* ssp. *andigena*[20], leaving 18 cultivars, from which we selected the ten most diverse to generate the pan-genome of European potato (Fig. 1b, Supplementary Table 2 and Supplementary Methods).

We sequenced the ten tetraploid genomes with 86.3 to 112.9 gigabases (Gb) of HiFi reads (26× to 36× coverage per haplotype) and generated initial contig assemblies of 2.3 to 2.7 Gb with contig N50 values of 2.3 to 3.6 megabases (Mb) using hifiasm[21] (Methods, Supplementary Methods and Supplementary Table 2). To address the assembly challenges of the tetraploid genomes[5–7], we developed a new pipeline integrating Hi-C reads into the assembly (Fig. 1c, Methods and Supplementary Methods). After validating the pipeline by reconstructing a previously assembled genome[6], we generated improved assemblies with 2.8 to 3.0 Gb in length and contig N50 values of 4.0 to 6.2 Mb. We scaffolded over 98.5% of the contigs of each assembly into four separate chromosome-level scaffolds for each of the 12 chromosomes (Fig. 1d, Extended Data Fig. 1, Supplementary Fig. 1 and Supplementary Tables 2 and 3). We annotated between 36,622 and 46,026 genes and 381 and 446 Mb of transposable elements (54.1% to 59.0% of the genome) in the 40 haploid genomes (Supplementary Table 4). The high variation in gene number was comparable to recent reports[6,7,22]. Assembly quality assessment[23] showed base quality values of approximately 45, greater than 97% completeness scores and greater than 95% BUSCO[24] completeness at both full assembly and gene levels (Extended Data Fig. 1, Methods and Supplementary Methods).

## Few but highly divergent haplotypes

On average, only 82% of each haplotype pair can be aligned against each other, even when high-resolution parameters were used (Fig. 2a–c, Supplementary Fig. 2, Methods and Supplementary Methods). The 18% unaligned regions were located primarily in pericentromeric regions measuring several Mb in size (Supplementary Figs. 3–14). Approximately 32% of the aligned regions were rearranged, such that on average only 56% of the genome was in synteny between any two haplotypes (Fig. 2a and Supplementary Fig. 2). The longest structural variations were inversions up to 37 Mb in size and were located primarily in pericentromeric regions (Fig. 2a and Supplementary Figs. 15 and 16). Some of the inversions were in the chromosome arms, such as an approximately 6-Mb inversion on chromosome 3, recently identified by a local depletion of recombination and linked to yellow tuber flesh[25]. Within the aligned regions, we found one variant site per 16 bp across all 40 haplotypes (Fig. 2b and Supplementary Figs. 17 and 18) and a pairwise nucleotide diversity ($\pi$) of 0.018, which corresponds to an average of one variant site per 56 bp between each pair of haplotypes (Fig. 2c and Supplementary Fig. 19). This exceptional sequence diversity is one of the highest reported in any domesticated crop so far[14,26], even though it varied considerably along the genome. Some regions showed significantly lower diversity than the genome-wide average (Extended Data Fig. 2). These low-diversity regions were located primarily in the pericentromeric regions of chromosomes 3, 6, 10 and 12 (Extended Data Fig. 3 and Supplementary Table 5).

By contrast, when analysing local haplotype diversity (that is, haplotype blocks measured in 10-kb windows along the chromosomes), we found only nine different haplotypes out of 40 potentially different haplotypes per haplotype block (Fig. 2d,e and Supplementary Fig. 20). Even in much larger windows, haplotype sharing remained high, and in some regions we even found shared haplotype blocks spanning multiple tens of Mb (Fig. 2f, Supplementary Fig. 21 and Extended

Data Fig. 4). On average, 18% of each haplotype pair was shared, with some haplotype pairs sharing up to 92% (Extended Data Fig. 5). In agreement with this extensive haplotype sharing, we also observed strong genome-wide linkage disequilibrium over long genomic distances ($\rho$ between $3.8 \times 10^{-7}$ and $1.8 \times 10^{-6}$; half-decay between 1.1 and 5.3 Mb), with $r^2 > 0.1$ extending between 5.7 and 27.4 Mb (Supplementary Table 6 and Supplementary Figs. 22 and 23). Notably, linkage disequilibrium was particularly high in regions with low genetic diversity and reduced haplotype numbers, consistent with reduced recombination in these regions (Extended Data Fig. 2). Taken together, despite an incredibly high sequence diversity, the genomes of European potatoes consist of only a few different haplotypes.

## Extensive admixture with wild species

The reduced haplotype diversity is probably reflecting recent population bottlenecks in potato history. However, the bottlenecks cannot account for the high sequence divergence between non-shared haplotypes. Recent reports suggested that high levels of admixture between wild and domesticated potato species may have caused the high diversity in potato[4,14,22]. To investigate this, we analysed the recently released genome sequences of 20 wild potato species[22] to identify introgression within each of the 40 European haplotypes (Fig. 3, Methods and Supplementary Methods).

We first constructed a phylogeny of wild potato species that revealed the three major clades of wild potato species[22]: Clade 1 + 2 (C1 + 2), Clade 3 (C3) and Clade 4 (C4), with C4 divided into predominantly southern South American (C4S) and northern South America (C4N) species (Fig. 3a). Regardless of the chromosomes used for the phylogeny, the wild potato species were consistently placed in the same phylogenetic positions, whereas the European haplotypes had different positions (Supplementary Fig. 24). Likewise, admixture analyses also revealed mixed ancestry of the European cultivars with shared components from multiple ancestral lineages, particularly with the C4S clade (at $K \geq 3$, where $K$ is the number of assumed ancestral populations; Fig. 3b and Supplementary Fig. 25). Together, this suggested substantial chromosome-specific introgressions in the cultivars.

We performed introgression tests (*D*-statistics and $f_4$ statistics) to identify introgressions of the C4S clade into the European cultivars (Fig. 3c–e, Supplementary Figs. 3–14 and 26–28 and Supplementary Methods). We did not test for introgressions between the species of clades C3 and C1 + 2 and the European haplotypes because of the low sequence similarity between them, which suggested that there are no introgressed regions. Likewise, we also did not test introgressions from clade C4N as the European cultivars were derived from this clade, making introgression tests unreliable.

The introgression tests showed strong support for numerous and long introgressions from the C4S clade across most haplotypes and chromosomes, which covered on average around 40% of the genome of a European haplotype (Fig. 3c–e and Supplementary Figs. 3–14 and 26–28). Species of C4S are usually found in southern South America, far from the Andean regions, which is the domestication centre of potato and the origin of the European haplotypes. Because our sampling of the European pan-genome is based on samples selected from the times of the onset of European breeding, these introgressions cannot be the product of modern potato breeding[15]. Instead, this suggests that these introgressions required human migration and transportation of plants during the times of domestication in South America.

The introgressed regions substantially overlapped regions with high sequence diversity (analysis of variance $F(1, 725) = 69.09$, $P = 4.6 \times 10^{-16}$; adjusted $R^2 = 0.31$), whereas regions with reduced diversity showed reduced amounts of introgressions (Fig. 3f and Extended Data Fig. 2). This proves that these introgressions introduced the high sequence diversity in the potato genome.

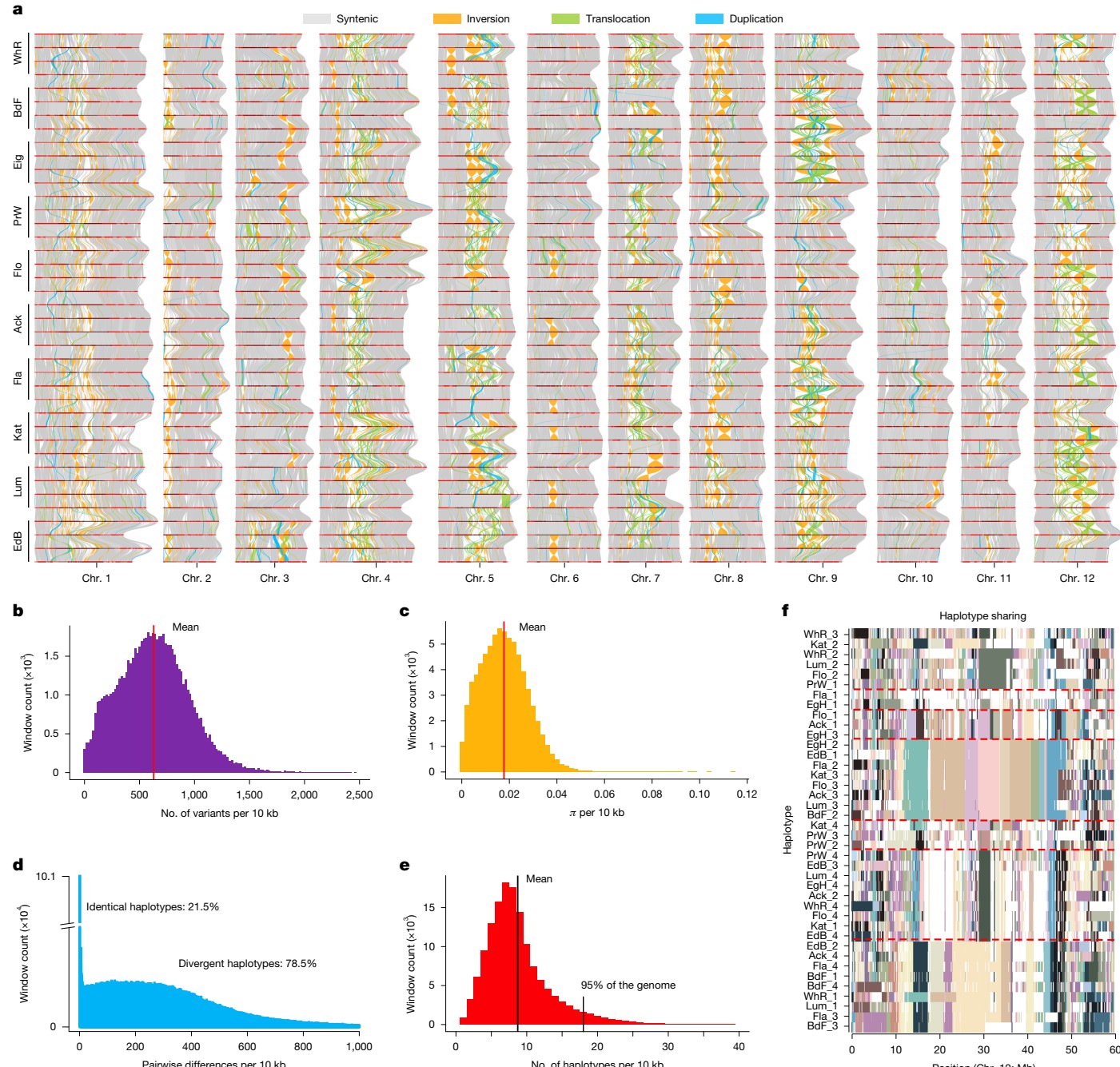

**Fig. 2 | Genetic and haplotype diversity in European potato. a**, Pairwise comparison of 40 haplotypes of all 12 chromosomes. Red lines, chromosome scaffolds; vertical lines, syntenic regions (grey), inversions (orange), duplications (cyan), translocations (green). **b**, Histogram of the number of variant sites across all 40 haplotypes measured in 10-kb windows. **c**, Histogram of pairwise nucleotide diversity $\pi$ (average: 0.018, indicated by the red dashed line) measured in 10-kb windows. **d**, Histogram of nucleotide diversity measured in 10-kb windows between all pairs of the 40 haplotypes across all 12 chromosomes. The diversity values revealed two clusters, that is, diverged haplotypes (78.5%)

with an average of 180 variations per 10 kb and identical haplotypes (21.5%) with less than 10 variations per 10 kb. **e**, Histogram of the number of unique haplotypes along the genome in 10-kb windows (average of 9 haplotypes indicated by the red dashed line). For 95% of the genome the number of unique haplotypes was below 18. **f**, Example of shared haplotypes across the 40 haplotypes for chromosome 12 (Supplementary Fig. 21). Colours indicate shared haplotypes. Haplotypes were clustered by sequence similarity. Long (tens of Mb) shared regions become apparent in the middle of the chromosome. Common haplotypes are highlighted with red dashed lines.

## 85% of the European haplotypes captured

The low number of distinct haplotypes in our sampling suggested that the overall haplotype space of European potato is also limited. To estimate the actual size of the pan-genome, we generated several pan-genome graphs with increasing numbers of haploid genomes (from 1 to 40)[27] (Fig. 4a, Methods and Supplementary Methods). With

each extra haplotype, the additional sequence contribution decreased. Curve fitting revealed that the full pan-genome size converges at 1.75 Gb. In turn, this implied that the 40 haplotypes (about 1.5 Gb in size) represent around 85% of the total variation of the European potato gene pool and that it would take only 24 further genomes to capture 95% of the entire variation, nearly completing the European potato pan-genome (Fig. 4a).

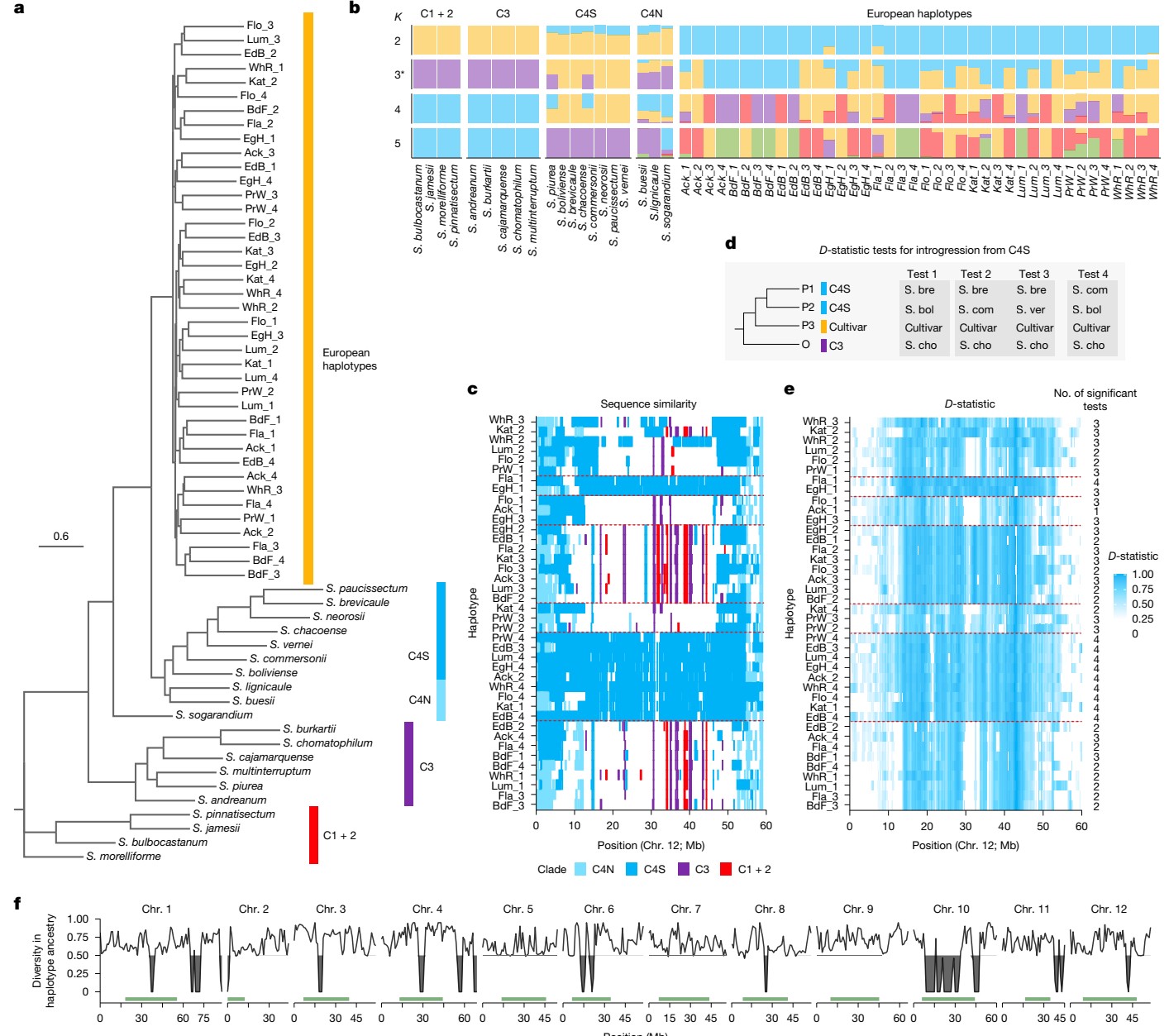

**Fig. 3 | Evidence of introgression and gene flow in European potato.**
**a**, Multispecies coalescence phylogeny of 40 haplotypes of cultivated potatoes and 20 genomes of wild potato species. Consensus tree of 850 maximum likelihood phylogenies for each 100-kb window across the genome (branch support values and individual consensus trees for each chromosome are shown in Supplementary Fig. 24). Main clades are highlighted. **b**, Admixture analysis across 20 wild diploid potato species and the 40 haplotypes of the potato cultivars (other chromosomes in Supplementary Fig. 25). Each colour represents a different ancestral population. The asterisk indicates the best-fitting *K*. **c**, Sequence similarity between wild species and cultivars along chromosome 12 (other chromosomes shown in Supplementary Fig. 26). Colours indicate the clade of the closest wild species. Common haplotypes are highlighted with

red dashed lines. **d**, Diagram illustrating the tests for introgression from the C4S clade. Each test involved four species (P1 to P3, and outgroup O), assessing introgression between P3 and either P1 or P2. The species names and clades are shown in panel **a**. **e**, Mean *D*-statistic results along chromosome 12 (Supplementary Fig. 27 for other chromosomes). Values are averaged across all tests within the clade. The total number of tests (maximum 4, see panel **d**) with statistically significant *f₄* statistics for the entire chromosome is indicated (individual *f₄* values in Supplementary Fig. 28). **f**, Diversity in haplotype ancestry along the genome from all 40 cultivar haplotypes. Green bars depict pericentromeric regions. Areas shaded in grey represent values below the 0.5 threshold, with the threshold level marked by a grey horizontal line along each chromosome.

A pan-genome with such a small size is expected to affect the gene space as well. To investigate this, we clustered the genes of all haplotypes using OrthoFinder[28] and defined 48,175 distinct gene families (Extended Data Fig. 6, Methods and Supplementary Methods). We repeated the pan-genome analysis using these gene families and found that the gene-based pan-genome was already saturated, implying that the 40 haplotypes include at least one representative of each gene

family in the European gene pool (Fig. 4b) and again evidencing the small overall size of the European pan-genome.

By contrast, the core-genome size (that is, the gene families shared by all genomes) had not yet converged and continued to decrease even after inclusion of all 40 haplotypes. In fact, the core-genome size was much smaller than what is usually shared between plant genomes, implying that the gene space of individual haplotypes is reduced.

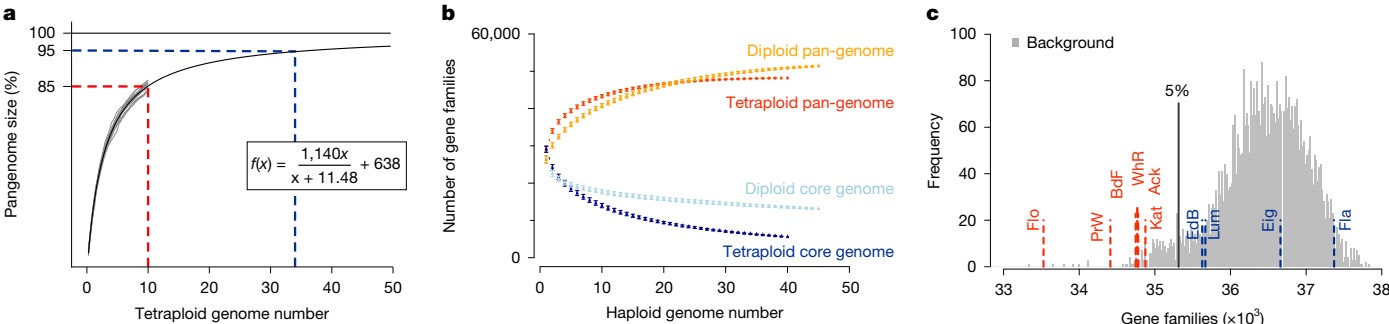

**Fig. 4 | Potato pan-genome analysis. a**, Estimating the total amount of sequence variation in the European potato. Pan-genome graphs with different amounts of haplotype genomes (that is, $x$ from 1 to 40) were used to fit a curve (that is, $f(x)$) that revealed the estimated size of the complete potato pan-genome. The unit of $f(x)$ in the given model is Mb; for example, $f(1) = 729.3$ Mb. The sequenced genomes cover 85% of this estimated variation. **b**, Pan- and core-genomes of the European potatoes (labelled as Tetraploid) compared with the pan- and core-genomes of diploid potato samples from recent studies[22] (labelled as Diploid). The number of genes shared by all haplotype genomes (core-genome: 5,602; 11.63%) in tetraploid potatoes was less than half of what was found in diploids (13,123; 25.5%) and did not converge. By contrast, the pan-genome reached a plateau indicating that the gene family space of European potatoes was greatly covered by the ten genomes analysed here, whereas the diploids had larger pan-genome size[22]. Data are presented as mean ± s.d., where mean values of gene families and error bars representing standard variations result from $n = 2,000$ random samplings during pan-genome construction (Supplementary Methods). **c**, The number of gene families in simulated and actual tetraploid genomes. The random background distribution was generated by repeatedly selecting four random, haploid genomes (of the 40) and calculating the number of gene families in them. Six of the ten genomes had significantly fewer gene families.

On average, individual haplotypes contained only 90.9% of the BUSCO genes. The reason for this is not clear from our analysis, but it is likely that the high load of deleterious mutations in tetraploid potato might have affected the gene space of individual haplotypes[29–31] (Supplementary Table 4). In turn, the tetraploid genome might compensate for the incomplete gene space of the individual haplotypes.

To test this, we performed random sampling of four haplotypes from the 40 haploid genomes to simulate random tetraploid genomes, for which we would expect that the real genomes have more genes than a random selection to compensate for the genes that are missing in individual haplotypes. Counterintuitively, however, six of the ten potato genomes showed significantly fewer gene families than random (Fig. 4c). This unexpected result might be simply explained by partial inbreeding, although this does outline how strongly the potato genome is affected by deleterious alleles.

## Phasing genomes with a haplotype graph

Phasing tetraploid potato genomes remains a challenging and time-intensive task. By contrast, resequencing genomes on the basis of the alignment of short reads against a single reference sequence is a well-established and straightforward alternative for many species. But as only a single reference sequence is used, resequencing is usually not powerful enough to reconstruct the highly divergent haplotypes of tetraploid potato genomes[32]. However, using multiple divergent haplotypes as a composite reference could facilitate the separation of the short reads during the alignments and in consequence this would allow the analysis of individual haplotypes of a tetraploid genome[33–35].

Modern elite potato cultivars are particularly well-suited for this approach. Because potato breeding is a slow process and only 5 to 15 sexual generations have been generated since the onset of modern breeding in Europe[36–39], their genomes probably retain very long haplotype blocks inherited from the original breeding founders, such as the ones we analysed here[16,40,41]. Therefore, using the founder genomes as a reference could enable the reconstruction of highly contiguous, phased genome sequences of modern elite cultivars using short reads alone.

To explore this, we converted the pan-genome into a genome graph, which we call a haplotype graph (Methods and Supplementary Methods). The nodes of the graph represent unique haplotype-specific sequences of 100 kb in size. Nodes were connected by edges if the respective haplotypes were linked in one of the assembled genomes. This approach preserves haplotype contiguity while simplifying the graph structure as all nearly identical 100-kb blocks are combined in individual nodes (Fig. 5a).

To use the haplotype graph for genome reconstruction, short reads of the genome of interest (that is, the query genome) need to be mapped to the graph. For this, all $k$-mers in the short reads are matched to the set of node-specific $k$-mers. After all $k$-mers in the reads are assigned to nodes, we can use the frequency of the $k$-mers assigned to individual nodes to estimate their copy number within the query genome. Connecting linked nodes with $k$-mer support enables the inference of continuous haplotype sequences within the query genome (Fig. 5b, Methods and Supplementary Methods). We refer to these inferred sequences as pseudo-contigs, which are similar to contigs of a conventional genome assembly, even though the sequences of the pseudo-contigs are taken from the connected nodes and not from the sequenced reads directly. We tested the haplotype graph with three different scenarios.

### Case 1. Reconstructing genomes included in the graph

To evaluate the performance of the haplotype graph, we generated a pseudo-genome assembly of 'White Rose', which is one of the cultivars that we used to create the haplotype graph. We used 85 Gb of short reads of 'White Rose' and matched their $k$-mers to the graph. Of 21,928 'White Rose' nodes in the haplotype graph, we found 17,331 nodes (recall = 79.0%) with the $k$-mers of the reads (or even 84.0% when ignoring the repetitive pericentromeres) (Fig. 5c and Supplementary Table 7). The recall of 'White Rose' haplotypes was unexpectedly high considering that this reconstruction used short reads alone. The predicted nodes had a precision of 83.5% (or 89.0% in the chromosome arms), where errors mostly occurred in regions where the actual 'White Rose' genome was not correctly represented in the graph (for example, owing to incomplete de novo assembly of 'White Rose') (Fig. 5d). In contrast to these types of errors, we did not observe more than 16 haplotype switch errors in the assembly, which is a common problem in the de novo assembly of tetraploid genomes. After connecting the supported nodes, we retrieved 2,883 pseudo-contigs which accounted for 71.7% of the tetraploid genome with a pseudo-contig N50 of 0.7 Mb (Fig. 5e,f). For the chromosome arms, the pseudo-contigs accounted for 84.9% of the genome with an N50 of 1.0 Mb.

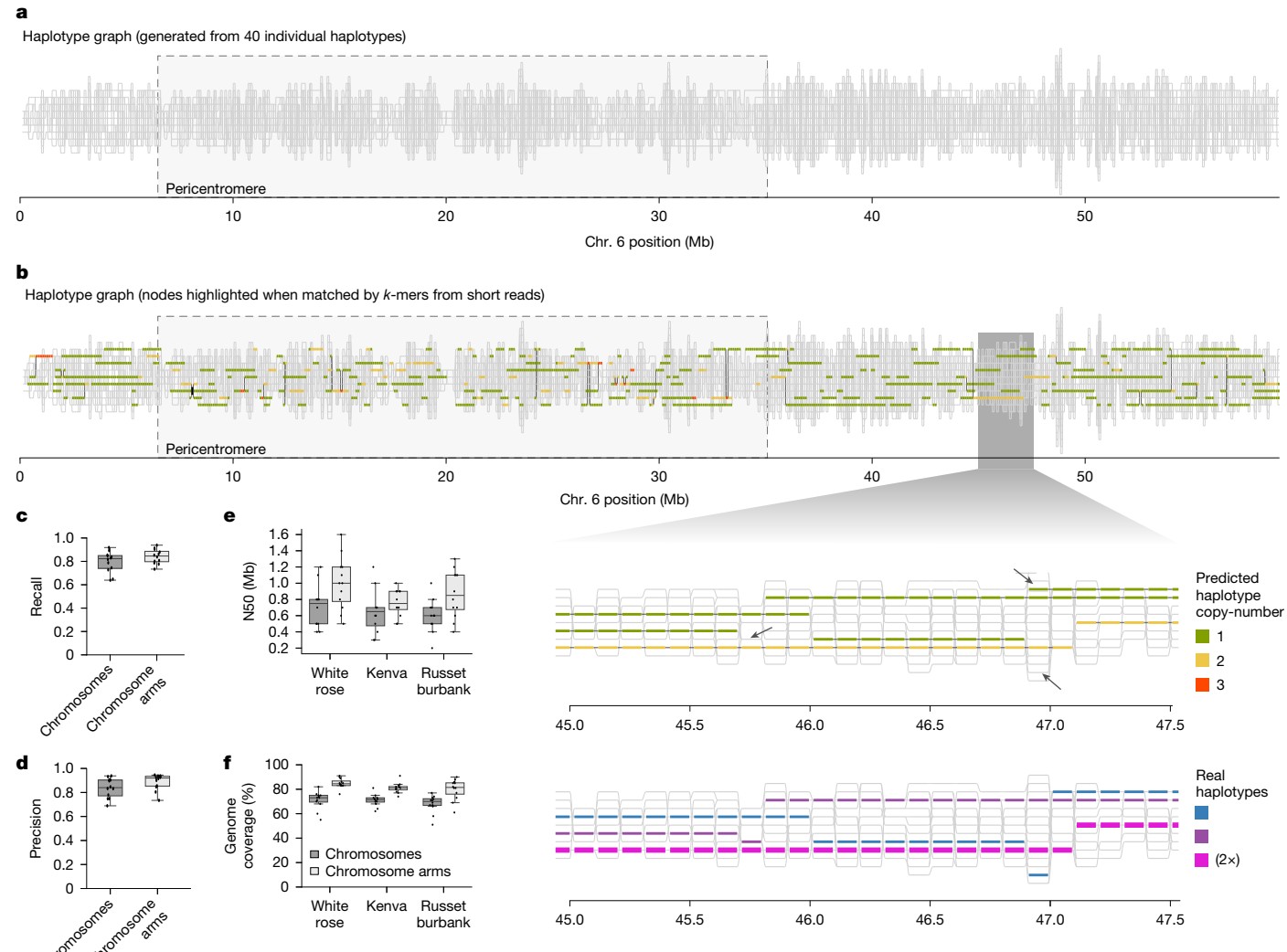

**Fig. 5 | Generating phased pseudo-genome assemblies with a haplotype graph. a**, A haplotype graph generated from the European potato pan-genome (here chromosome 6 is shown). Each horizontal line visualizes a node representing shared haplotypes. Divergent haplotypes are assigned to different nodes. **b**, Mapping the *k*-mers of a sequenced sample to the graph. The nodes of the graph are highlighted in green, yellow or red if the unique *k*-mers of the nodes are supported by the *k*-mers in a read dataset of the sequenced 'Kenva' genome. Depending on the amount of *k*-mers, it is estimated that the haplotypes of the respective nodes are present one (green), two (yellow) or three (red) times in the genome of 'Kenva'. Lower panels, magnification of a specific region in the genome. The three black arrows show small errors in the reconstruction of the

haplotypes. The real haplotypes are shown in the lower panel. The haplotypes shown in pink are represented twice in the genome of 'Kenva'. **c**–**f**, Performance estimates of the haplotype-graph-based pseudo-genome assembly. **c**, Proportion of expected haplotype nodes recovered in the pseudo-genome assembly for 'White Rose' (recall). **d**, Proportion of predicted nodes correctly matching the 'White Rose' genome (precision). **e**, N50 values of pseudo-genome assemblies for each cultivar. **f**, Percentage of the genome assembled into pseudo-contigs (coverage) for each cultivar. Panels **c**–**f** show results both including and excluding pericentromeric regions. For these plots, *n* = 12. The boxes show the 25th quantile, median and 75th quantile. Whiskers extend to points within 1.5× the inter-quantile range. Points outside this are plotted as outliers.

## Case 2. Reconstructing genomes derived from the haplotypes in the graph

We further sequenced the genome of 'Kenva' (Supplementary Methods), a cultivar that was derived from three of the genomes that were included in the graph (resulting from a cross between 'Flava' and 'Calrose' (a hybrid of 'Ackersegen' and 'Katahdin')), whereas the genome of 'Kenva' itself was not included in the graph. This allowed simulating the (idealized) scenario of a complete haplotype graph that includes all haplotypes segregating in the European potatoes.

Using 100 Gb of short reads of 'Kenva', we generated 3,114 pseudo-contigs covering 70.9% of the genome with an average N50 of 0.6 Mb (in chromosome arms: genome coverage = 81.1%; N50 = 0.8 Mb) (Fig. 5e,f). The N50 values were lower than what we achieved with 'White Rose', probably owing to a combination of reasons, including recombinant haplotypes in the genome of 'Kenva'. Overall, however, the contiguity

of the pseudo-contigs was still very high, considering that the assembly was built from short reads only (Supplementary Table 7).

## Case 3. Reconstructing unknown genomes

Finally, we used the haplotype graph to generate a pseudo-genome assembly of 'Russet Burbank', a commercial elite variety used for the production of French fries, for which a genome assembly has not yet been publicly released.

Using 67 Gb of short-read data of 'Russet Burbank' mapped to the haplotype graph, we generated a pseudo-genome assembly that included 2,769 pseudo-contigs of 2.03 Gb. The pseudo-assembly covered 68% of the estimated genome (79.3% of the chromosome arms), with an N50 of 0.6 Mb (Fig. 5e,f and Supplementary Tables 2 and 7).

To evaluate the pseudo-genome assembly of 'Russet Burbank', we additionally generated a phased de novo assembly of 'Russet Burbank',

following the same procedure as for the ten other cultivars (Methods and Supplementary Methods).

Comparing the pseudo-contigs with the de novo assembly showed that approximately 87% (1.75 Gb) of the pseudo-contigs (almost) completely aligned (more than 95% of the pseudo-contig length) to a single haplotype of the de novo assembly, including a very long pseudo-contig measuring 9.9 Mb (Supplementary Fig. 29). The remaining approximately 13% aligned to more than one haplotype, indicating that these were probably chimeras of different haplotypes. In addition, some of the pseudo-contigs included dense clusters of sequence differences from the de novo assembly, which probably results from haplotypes in the 'Russet Burbank' genome that were not yet represented in the graph. Although this might be a limitation of the haplotype graph (as long as the underlying pan-genome is not completed), pseudo-contigs can be polished to correct some of these differences. Using a more complete pan-genome will reduce the number of pseudo-genome assembly errors, and carries the promise to enable phased and chromosome-level assemblies of potato genomes using short reads only.

## Discussion

Here we present a haplotype-resolved pan-genome of tetraploid European potato, which includes more than 85% of the genetic variation of potato within Europe. The high sequence diversity between the haplotypes could be explained by introgressions from wild species during domestication covering an extremely high proportion (about 40%) of the cultivated potato genome. By contrast, haplotype diversity was strikingly reduced, which was in agreement with multiple bottlenecks in the history of European potato. This low level of haplotype diversity might represent a call to action for widening the haplotype space of European potatoes. However, including non-adapted or non-elite material into the modern gene pool is not straightforward. Foreign material might lack quality traits or bring in new deleterious alleles that would need to be purged from the genomes. Instead, genome editing might become an alternative way to advance the variation of the limited gene pool, either by introducing new beneficial alleles or by purging linked deleterious alleles. The recent decision of the European Union to agree to the use of genetic engineering in agriculture might promote such efforts in the European potato breeding programmes.

But the limited haplotype space also offers opportunities. In a proof of concept, we illustrated how the pan-genome can be converted into a genome graph (called haplotype graph) that can be used to reconstruct the haplotypes of a modern cultivar using cost-efficient short-read data only. The development of genome graphs is an active research field[27,42–45], and recently pan-genomes of several crop species were constructed[46–48]; however, most of the current genome-graph tools would not support haplotype phasing of autotetraploid genomes. The pseudo-genome assemblies that we generated with short reads and the haplotype graph featured N50 values close to the values of de novo assemblies with long reads. The assemblies covered around 80% of the genome, which is an enormous fraction considering the repetitiveness of the potato genome, even though the precision of the assemblies was around 90% which implied some falsely included haplotype sequences as well. As many of these errors were related to missing information in the haplotype graph, we expect that the precision of the pseudo-assemblies will improve when more genomes are included in the haplotype graph. Such cost-efficient pseudo-genome assemblies will open the door for the broad exploration of large breeding panels, such as the more than 1,600 European potato cultivars that are at present registered for market access in the European Union[49].

The haplotype graph method will continue to improve until the pan-genome captures the entire genetic variation in European potato. We estimated that it would require only 24 further genome assemblies to capture 95% of the haplotype space of European potatoes. Although this would still be feasible, it would require yet another 145 random

genomes to reconstruct 99% of the haplotype space (as progressively less variation is added by further genomes). But instead of sequencing random genomes, it is more efficient to target cultivars that are known to carry haplotypes that were not included in the pan-genome so far. This is specifically important for integrating introgressions that were made by breeders in past decades, which have not been covered by our assemblies at all. Using genotype or pedigree data to select cultivars for sequencing would markedly reduce the number of genomes needed to complete the pan-genome—a daunting task, which so far has not been attempted for any species at all. However, considering the very low number of different haplotypes in European potato, there is no reason why the generation of a truly completed pan-genome would not be possible within the next few years.

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

## Methods

Details are provided in the Supplementary Information.

### Plant material collection

The potato cultivars (Supplementary Table 2) were clonally propagated and grown on Murashige–Skoog medium for 3–4 weeks at Max Planck Institute for Plant Breeding Research (MPIPZ, Germany). Plantlets were transferred to soil in 7 × 7-cm$^2$ pots and grown in a Percival growth chamber for 2–3 weeks. Afterwards, the plants were transferred to 1-litre pots and grown until flowering. The plants were grown in long-day conditions (16-h light, 8-h night cycle) at 22 °C.

### PacBio HiFi long-read sequencing

High-molecular-weight (HMW) DNA was isolated from 1.5 g of material with a NucleoBond HMW DNA kit (Macherey Nagel). Quality was assessed with a FEMTOpulse device (Agilent) and quantity measured by fluorometry by Quantus (Promega). A PacBio HiFi library was prepared according to the manual 'Procedure & Checklist – Preparing HiFi SMRTbell Libraries using SMRTbell Express Template Prep Kit 2.0' (https://www.pacb.com/wp-content/uploads/Procedure-Checklist-Preparing-HiFi-SMRTbell-Libraries-using-SMRTbell-Express-Template-Prep-Kit-2.0.pdf) with initial DNA fragmentation by g-Tubes (Covaris) and final library size binning by SageELF (Sage Science). Size distribution was again controlled by FEMTOpulse (Agilent). Size-selected libraries were sequenced on a Sequel II device at Max Planck Genome-centre Cologne (MP-GC) with Binding kit 2.0 and Sequel II Sequencing Kit 2.0 for 30 h. (Read statistics are provided in Supplementary Table 2).

### Omni-C sequencing

For each cultivar, an aliquot of HMW DNA was extracted from fresh leaves and used for a Dovetail Omni-C library created at MP-GC using the Omni-C Kit. The libraries were sent to BGI, Hongkong (China) on dry ice, where they were sequenced on a DNBseq platform (Supplementary Table 2). DNBseq reads were filtered with SOAPnuke filter with parameters '-T 4 -l 20 -q 0.2 -n 0.001 -4 100' (v.2.1.7)[50].

### Whole-genome short sequencing

**Low-coverage sequencing.** Fresh leaves were sampled from 19 different cultivars and DNA was extracted with the Nucleo Mag Plant kit (Macherey Nagel), followed by Nextera LITE DNA preparation[5]. The libraries were sent to BGI, Hongkong (China) on dry ice, where they were sequenced on a DNBseq platform (Supplementary Table 2). Principal component analysis was used to analyze these sequencing samples to select the ten most diverse cultivars for pan-genome construction (Supplementary Methods). Blinding and randomization were not used.

**High-coverage sequencing.** Fresh leaves were sampled from ten selected cultivars and DNA was extracted with the Nucleo Mag Plant kit (Macherey Nagel). Genomic DNA was sent to BGI, Hongkong (China) on dry ice and whole-genome shotgun libraries were prepared according to the standard protocol of BGI, where they were sequenced on a T7 DNBseq platform (Supplementary Table 2). Cultivar 'Russet Burbank' was processed in the same way. DNBseq reads were filtered with SOAP-nuke filter with parameters '-T 4 -l 20 -q 0.2 -n 0.001 -4 100' (v.2.1.7)[50].

### Phased assembly of autotetraploid genomes

Individual genome sizes were estimated using Jellyfish[51] (v.2.2.10) and findGSE[52] (v.1.0). Initial genome assemblies were generated with hifiasm[21] (v.0.7) with default settings, and contigs with low sequencing support were purged. The contigs were classified on the basis of sequencing coverage (using samtools[53] depth function (v.1.9)): according to the average sequencing depth per haplotype $d$, contigs with [0, 1.5 $d$], [1.5 $d$ + 1, 2.5 $d$], [2.5 $d$ + 1, 3.5 $d$], [3.5 $d$, 4.5 $d$] and [4.5 $d$ + 1, infinite] were determined as haplotig, diplotig, triplotig, tetraplotig and replotig (Supplementary Fig. 30). Haplotigs were phased using Hi-C data to create haplotype-specific groups. Diplotigs, triplotigs and tetraplotigs were phased to two, three and four of the groups using code developed in this study (https://github.com/HeQSun/tetraDecoder). Next, HiFi reads were re-aligned to the contigs in each group, allowing a further, independent assembly of each haplotype. The resulting contigs were scaffolded with Hi-C data. Finally, Hi-C contact maps were built up with all four haplotypes of each chromosome using the ALLHiC[54] package (v.0.9.13), and mis-phased regions were manually searched and corrected.

### Gene annotation and assembly evaluation

The contig-level assemblies were annotated using BRAKER (v.2.1.6)[55–57]. The GFF files of the BRAKER1 (refs. 58,59) and BRAKER2 (refs. 60–64) workflows were combined using TSEBRA[65] (v.1.0.3), representing the final gene annotations. Gene annotations of each contig assembly were transferred to their respective final chromosome-level assemblies using liftoff (v.1.6.2)[66]. Base quality and sequence-level completeness of the genome assemblies were assessed using Merqury (v.1.3)[23], and gene set completeness was evaluated using BUSCO (v.5.2.2)[24].

### Comparison of haplotype sequences

For each chromosome, each of the 40 haplotype-specific sequences were aligned to each other using nucmer3 (v.3.1)[67] with options '--maxmatch -c 100 -l 80 -b 500'. The resulting files were processed using delta-filter with options '-m -i 85 -l 200', and further with show-coords with option '-THrd', which resulted in coordinate files that were provided to SyRI (v.1.6)[68] to call single-nucleotide polymorphisms (SNPs), structural variations and syntenic regions between all paired sequences. Visualization of the chromosome-level comparisons was performed with a customized version of plotsr[69] (https://github.com/schneebergerlab/plotsr/tree/chr_objects).

### Analysis of genetic diversity

Haplotype-specific sequences of each cultivar were aligned to the reference genome double monoploid (DM) 1-3 516 R44 using nucmer3 (v.3.1)[67,70]. SyRI (v.1.6)[68] was used to call SNPs, structural variations and syntenic regions. The distribution of structural variation across the genome was determined using Msyd (v.1.0) (https://github.com/schneebergerlab/msyd).

Genetic variants of pairwise haplotype–DM comparisons were merged into a single genotype table. The genotype table was used to calculate minor allele frequency, pairwise nucleotide diversity ($\pi$)[71], Watterson theta ($\theta_w$)[72] and linkage disequilibrium, as well as to cluster the haplotypes (Supplementary Methods).

### Introgression identification

Whole-genome sequencing reads of 20 wild potato species were aligned to the DM reference genome[70] and cultivar haplotypes using minimap2 (v.2.20-r1061)[73]. Variant calling was performed using DeepVariant (v.1.4.0)[74]. The variants were merged into a unified dataset. Read depth was calculated across cultivar genomes using Mosdepth (v.0.3.1)[75] in 100-kb windows and used to evaluate potential introgressions.

Phylogenetic relationships were analysed in 100-kb windows by constructing maximum likelihood trees with IQ-TREE (v.2.1.2)[76] under the general-time-reversible (GTR) model with 1,000 bootstrap replicates. Consensus trees for each chromosome and the whole genome were generated using ASTRAL (v.5.7.8)[77]. Admixture analysis was conducted with ADMIXTURE (v.1.3.0)[78] for $K$ values from 2 to 10. In addition, $D$-statistics (ABBA-BABA)[79] and $f_4$ statistics[80,81] were calculated using Dsuite (v.0.5)[82] to detect introgressions. Tests for introgressions were performed per chromosome and in 200-kb sliding windows.

## Pan-genome construction

The pan-genome was initialized with a single haplotype. Further haplotypes were iteratively incorporated using alignments against the haplotypes that were already included in the pan-genome using minigraph (v.0.20-r55966)[27] with parameters '-cxggs -t 20'. A model, $y = a_1 \times x/(x + a_2) + a_3$, fitting the increasing pattern of the pan-genome size was constructed, for which the parameters were optimized using the BFGS method in R 4.3.0 (ref. 83).

A gene-level pan-genome was constructed using all genes in the 40 haploid genomes, which were first clustered with OrthoFinder (v.2.5.5)[28], diamond (v.2.0.13)[84] and Blast (v.2.12.0+)[85]. Core, softcore and dispensable genes were defined as shared by 40, 37–39 and 2–36 of the haplotype genomes. Private genes were specific to individual haplotypes. The pan-genome and the core-genome distributions were built on the basis of the OrthoFinder gene families, for which up to 2,000 random samplings from the 40 haploid genomes were performed for each sample size (from 2 to 40).

## Haplotype phasing with haplotype graph

**Haplotype graph construction.** The pan-genome was transformed into a haplotype graph with the alignments of the 40 haplotype genomes to the DM reference using minimap2 (ref. 73). Genomic variants were identified using SyRI[68]. Specifically, the reference genome coordinates were binned into non-overlapping 100-kb windows. Haplotypes in each window were clustered together if the edit distance between their SNP profiles was less than 10% of the number of SNPs (that is, for any two haplotype instances assigned to the same cluster, they show less than 10 SNP differences per 100 SNPs to each other). In the haplotype graph, each cluster of haplotypes is then represented with one node. The nodes in adjacent windows were connected by edges if they were linked in any of the contributing haplotypes. For each node, we identified marker $k$-mers ($k = 51$) that were (1) in all the contributing haplotypes, (2) in regions being in synteny with DM and (3) unique to that node.

**Phasing query genomes using the haplotype graph.** Using short reads of a query genome, marker $k$-mers were extracted using Jellyfish[51] (v.2.2.10). The probability of a $k$-mer representing zero, one, two, three or four haplotypes was estimated with a Gaussian mixture model. On that basis, an iterative process of expectation maximization was used to classify nodes in the haplotype graph as representing zero, one, two, three or four copies in the sample, which would not finish until all nodes converged (minimum step size < 0.001) or 100 iterations were reached. Nodes with a non-zero copy number were then heuristically connected to form pseudo-contigs.

## Reporting summary

Further information on research design is available in the Nature Portfolio Reporting Summary linked to this article.

## Data availability

The haplotype-resolved genome assemblies and annotation of the 11 genomes are available at Zenodo (version 2: https://doi.org/10.5281/zenodo.10617012; version 2.1: https://doi.org/10.5281/zenodo.14053896)[86]. All sequencing data generated in this study are available at NCBI under BioProject PRJNA1074690. Genome long-read sequencing data of 20 wild species were retrieved from NCBI under BioProject PRJNA754534, and that of 'Otava' were retrieved from NCBI under BioProject PRJNA726019. Reference data of DM 1-3 516 R44 (v.6.1) were downloaded from SpudDB (https://spuddb.uga.edu/download.shtml) and reference data of dAg1_v1.0 are available at Figshare (https://doi.org/10.6084/m9.figshare.14604780)[87]. RNA-seq data were retrieved from the National Genomics Data Center (https://bigd.big.ac.cn) under project PRJCA007997, protein sequences of tomato (ITAG4.0) were downloaded from https://solgenomics.net/ and the hierarchical catalogue of orthologs were downloaded from https://www.orthodb.org/ (orthoDBv10). Source data are provided with this paper.

## Code availability

Custom scripts developed in this project are available at GitHub. For phased tetraploid genome assembly and pan-genome analysis: https://github.com/HeQSun/tetraDecoder; for phasing genomes using haplotype graph: https://github.com/schneebergerlab/pantohap and https://github.com/schneebergerlab/haplotype_threading. All codes and scripts are also available at Zenodo (https://doi.org/10.5281/zenodo.14786484)[88].

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

**Acknowledgements** This work was funded by the Research Fund for Young Talent Plans of Xi'an Jiaotong University (grant no. KZ6JO07) (H.S.), the National Natural Science Foundation of China (grant no. GYKPO34) (H.S.), the Deutsche Forschungsgemeinschaft (DFG; German Research Foundation) under Germany's Excellence Strategy (grant no. EXC 2048/1–390686111 (C.I.D. and K.S.) and grant no. SCHN1257/15-1 (K.S.)) and the European Research Council (ERC) grants 'INTERACT' (no. 802629) and 'BYTE2BITE' (no. 101124694) (both K.S.). HPC computing was performed on the BioHPC hosted at Leibniz Rechenzentrum Munich funded by the German Research Foundation (grant no. INST 86/2050-1 FUGG). In addition, we thank S. Huang (AGIS, Shenzhen, China) for helpful suggestions on the manuscript; P. Flood (Aardaia, the Netherlands), Q. Lian (Max Planck Institute for Plant Breeding Research, Germany) and J. Wolf (LMU Munich, Germany) for helpful discussions; and P. Włodzimierz and I. Henderson (University of Cambridge, UK) for sharing software before release.

**Author contributions** H.S., S.T. and K.S. developed the project. J.A.C., B.W., C.S. and B.H. performed plant work and generated data. H.S., S.T., C.I.D., M.G., R.Y.W., L.C.B., X.D., A.K. and K.S. performed all data analysis. H.S., C.I.D., R.C.B.H., H.J.v.E. and K.J.D. analysed pedigree data and selected the pan-genome samples. H.S., S.T. and K.S. wrote the manuscript with input from all authors. All authors read and approved the final manuscript. C.I.D. and M.G. contributed equally to this work.

**Funding** Open access funding provided by Ludwig-Maximilians-Universität München.

**Competing interests** The authors declare no competing interests.

**Additional information**
**Correspondence and requests for materials** should be addressed to Korbinian Schneeberger.

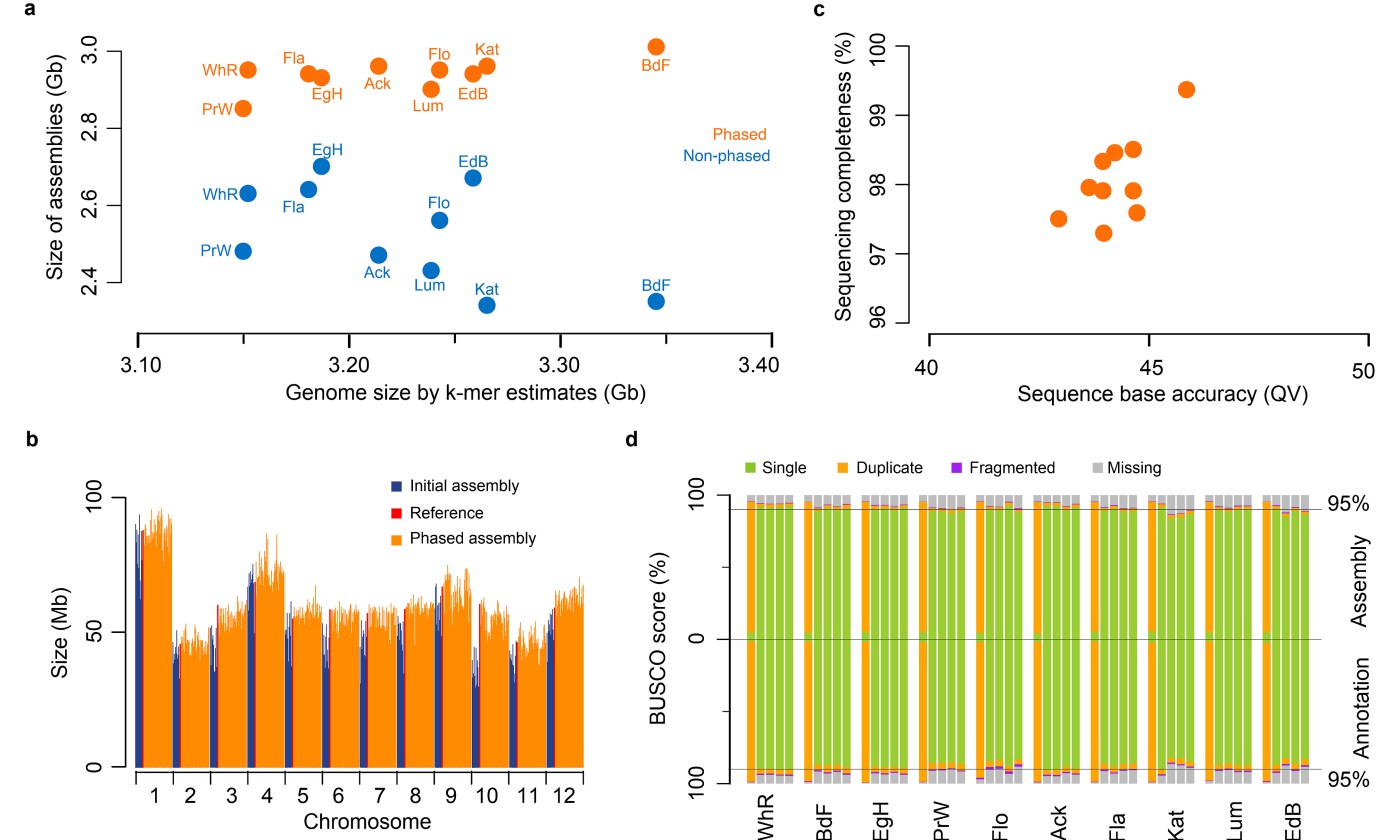

**Extended Data Fig. 1 | Evaluations of genome assemblies. a**. Assembly size before and after haplotype phasing of ten potato varieties, as compared with *k*-mer-based genome size estimates. **b**. Total contig size of chromosomes of initial and phased assemblies as compared with the reference genome. As a comparison, for example, chromosomes 3,6,7,10 after phasing (in orange) were clearly more complete than those before phasing (in dark blue). Note, as the total size of initial contigs aligned to each reference chromosome included contigs mixed from four haplotypes, the average size (i.e., the total size divided by four) was used to represent the initial assembly size of a haplotype chromosome. **c**. Base accuracy and completeness of ten tetraploid genome assemblies calculated by *Merqury*[23]. **d**. BUSCO[24] evaluation of 40 haploid genome assemblies, based on the genome assemblies and the protein-coding gene annotations.

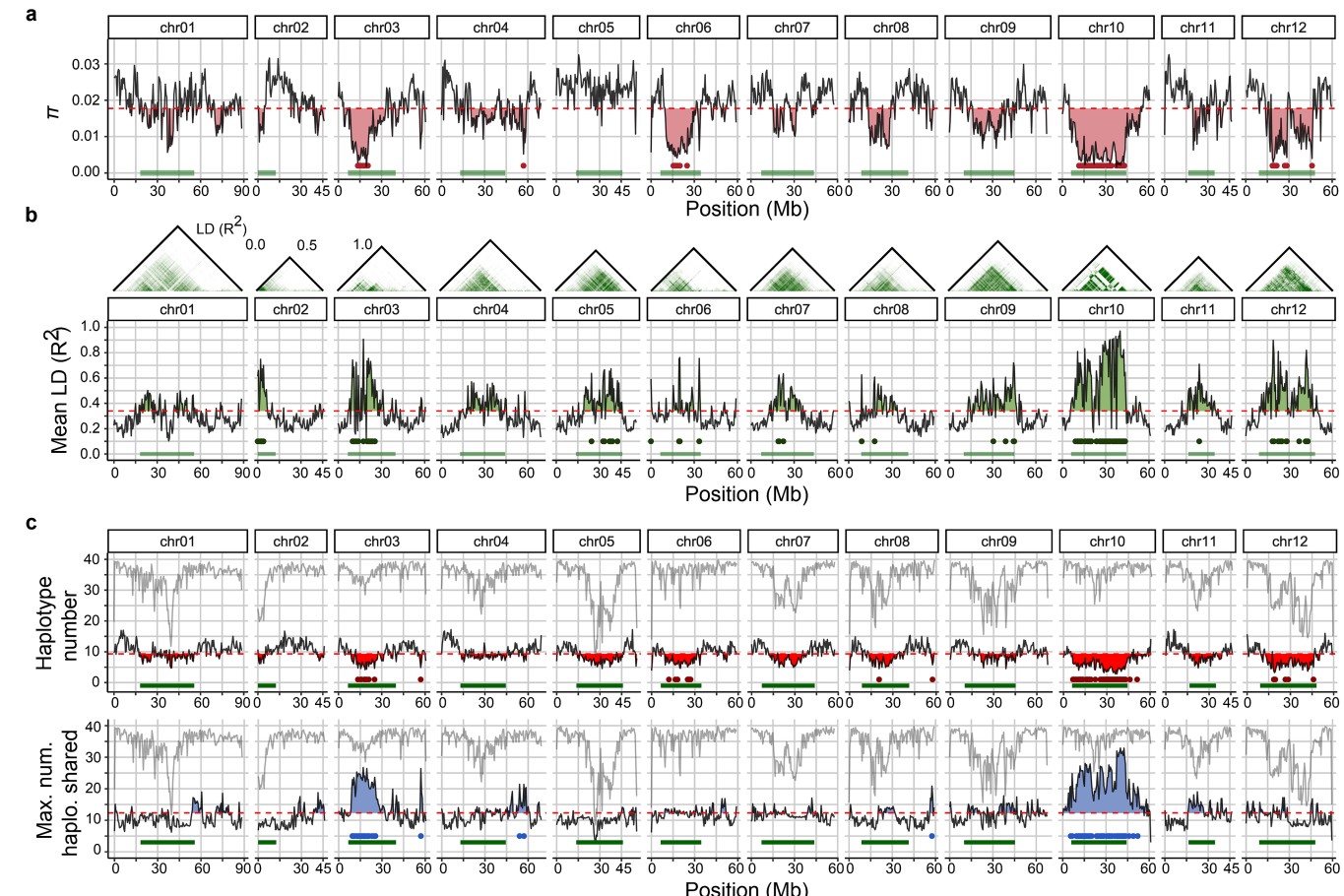

**Extended Data Fig. 2 | Diversity and footprints of selection in the European potato. a**. Average pairwise nucleotide diversity π (average at 0.018 indicated by the red dash line) along the genome in 10 kb windows. Red dots indicate regions with the 5% lowest π values (below 0.003). Values calculated per 10 kb window from 40 haplotypes. **b**. Linkage disequilibrium (LD) within the 40 haplotypes across each of the 12 chromosomes. LD measured as $R^2$ is shown as a heatmap throughout all chromosomes. The bottom panel shows the average $R^2$ within windows of 200 kb. Dots in dark green indicate the windows with 5% most extreme values in the genome. **c**. Top panel: Number of different haplotypes in 10 kb windows (averaged across 20 windows) out of 40. The dashed line in red shows a genome-wide average of only nine distinct haplotypes per region. Bottom panel: Maximum haplotype allele frequency along the genome. Each window reports the number of haplotypes carrying the most frequent haplotype. Red and blue dots indicate the windows with 5% most extreme values. Green lines: pericentromeric regions. Light gray lines: number of aligned haplotypes to the reference sequence.

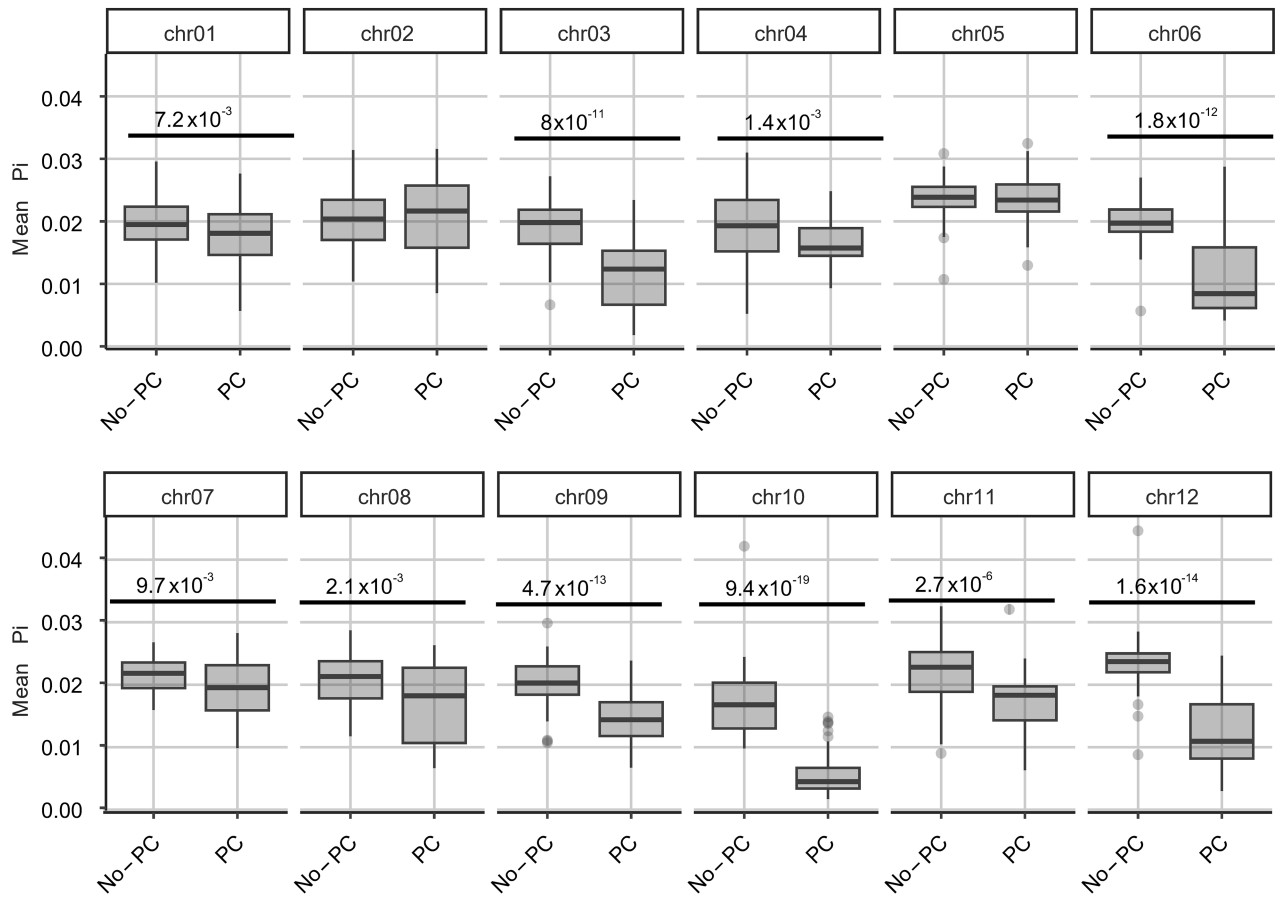

Genomic region: pericentromeric (PC) or non-pericentromeric (No-PC)

**Extended Data Fig. 3 | Mean estimate of genetic diversity ($\pi$) for pericentromeric (PC) and non-peri centromeric regions (No-PC) across chromosomes (chr01-12).** Box plots represent the median (center line) from 40 haplotypes, interquartile range (bounds of the box), and minima/maxima (whiskers), excluding outliers (points), which are defined as values beyond 1.5× the interquartile range. Statistical significance was determined using a two-sided Wilcoxon rank-sum test, with $p$-values adjusted for multiple comparisons using the Benjamini-Hochberg (BH) method. Sample sizes (n) in

number of 500 kb windows for pericentromeric (PC) and non-pericentromeric (No-PC) regions per chromosome: chr01 (75, 103), chr02 (24, 69), chr03 (66, 56), chr04 (62, 77), chr05 (64, 48), chr06 (57, 62), chr07 (73, 43), chr08 (65, 54), chr09 (71, 65), chr10 (77, 46), chr11 (37, 57), chr12 (78, 42). For all chromosomes combined: PC = 749, No-PC = 722. Genetic diversity in pericentromeric regions was consistently lower than in non-pericentromeric regions, with the exception of chromosome 1 and 2 where no significant differences were found (p > 0.05). Significant differences are indicated with p-values.

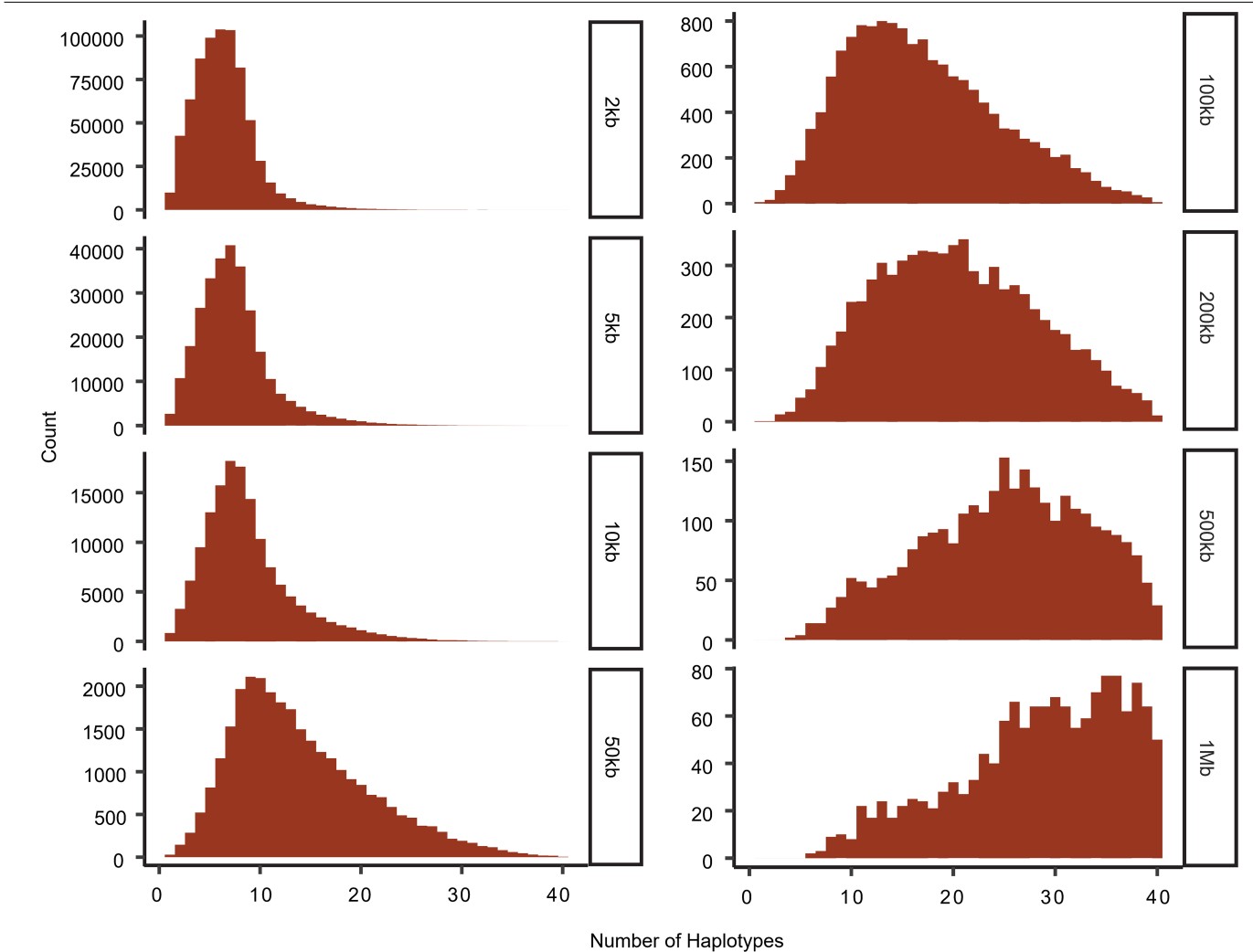

**Extended Data Fig. 4 | Distribution of haplotype numbers with different window sizes.** Histograms of the number of different haplotypes measured in windows with different sizes ranging from 2 kb to 1 Mb. In all panels values were calculated from all 40 haplotypes.

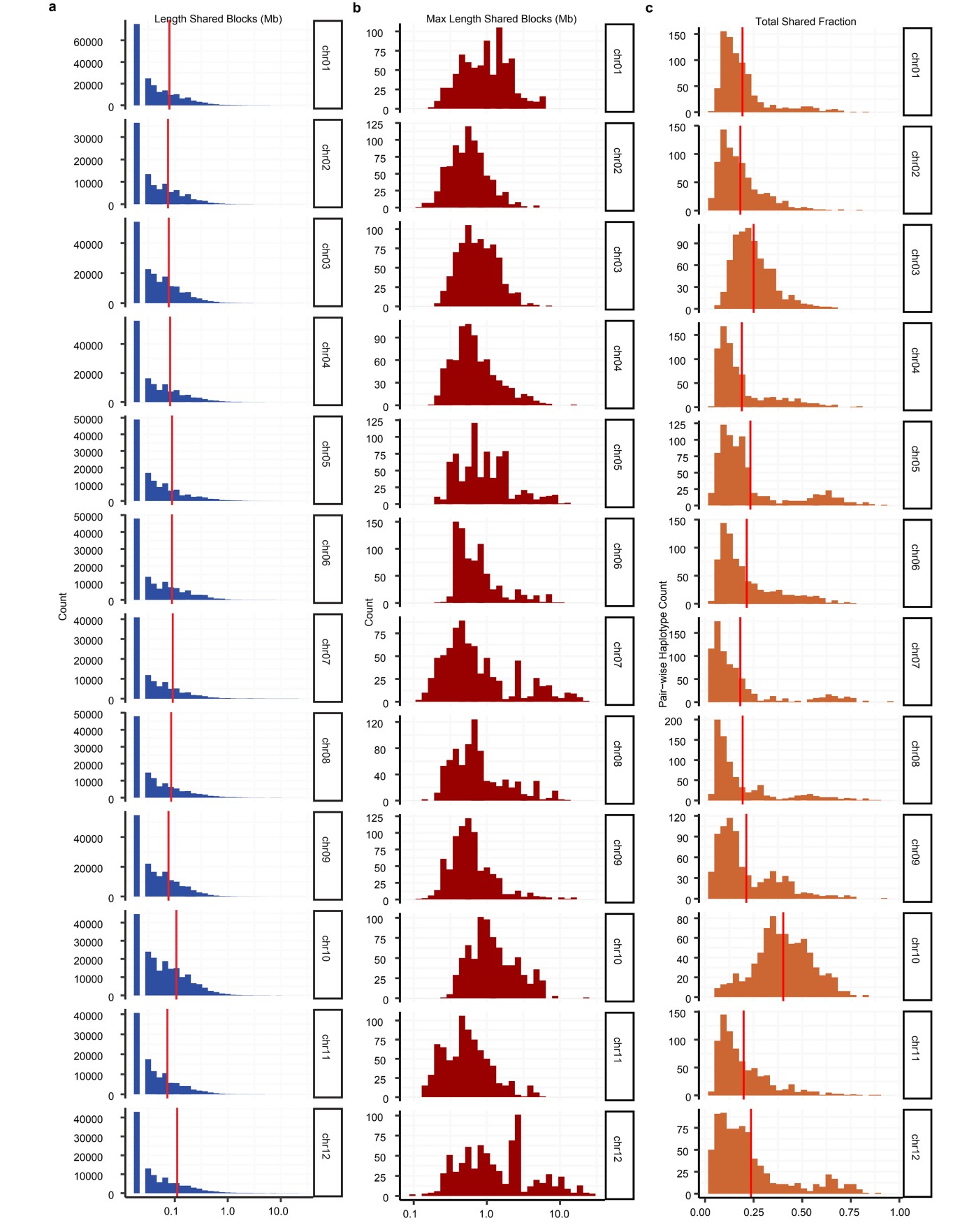

**Extended Data Fig. 5 | Haplotype sharing. a.** Histogram of shared block sizes in all pairwise haplotypes comparisons. **b.** Histogram of the maximum shared block size per pairwise haplotype comparison. **c.** Histogram of the proportion of shared haplotype sequence in all pairwise haplotype comparison. Panels are split by chromosome. Mean values are indicated with a red vertical line. In all panels values were calculated from all 40 haplotypes.

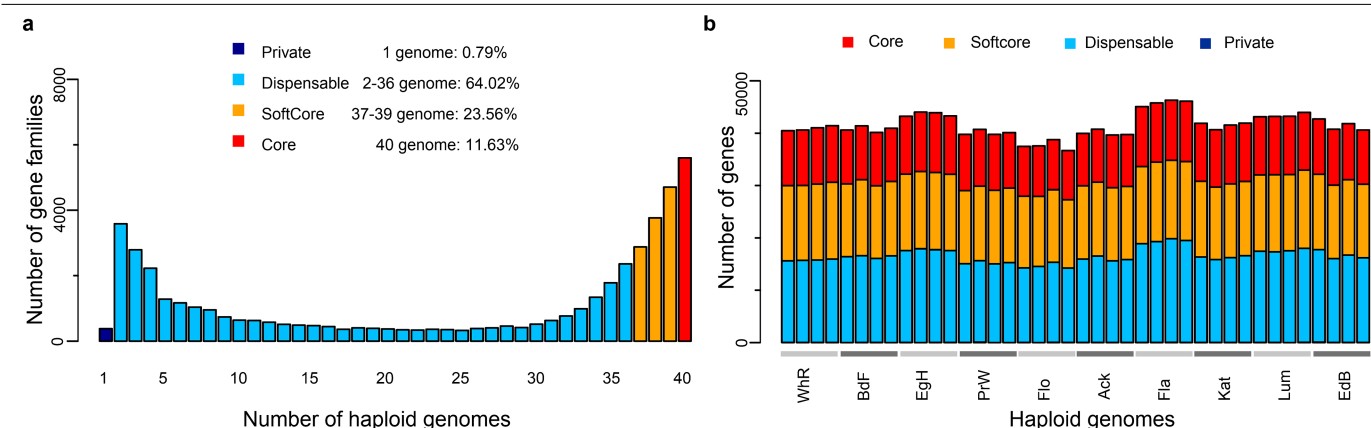

**Extended Data Fig. 6 | Gene (family) number and gene sharing. a.** Distribution of private, dispensable, softcore and core gene families. Private (only found within a single haploid genome), dispensable (shared by 2–36 haploid genomes), softcore (shared by 37–39 haploid genomes) and core (shared by all 40 haploid genomes) gene families represented 0.79%, 64.02%, 23.56% and 11.63% of all gene families, respectively. **b.** Distribution of private, dispensable, softcore and core genes in single haploid genomes, as characterized in a. Nearly no private genes were found within individual haploid genomes, while dispensable genes were highly abundant. This analysis involved 40 haploid genomes in total.

# Reporting Summary

## Statistics

For all statistical analyses, confirm that the following items are present in the figure legend, table legend, main text, or Methods section.

| n/a | Confirmed | |
|---|---|---|
| ☐ | ☒ | The exact sample size (*n*) for each experimental group/condition, given as a discrete number and unit of measurement |
| ☒ | ☐ | A statement on whether measurements were taken from distinct samples or whether the same sample was measured repeatedly |
| ☐ | ☒ | The statistical test(s) used AND whether they are one- or two-sided<br>*Only common tests should be described solely by name; describe more complex techniques in the Methods section.* |
| ☒ | ☐ | A description of all covariates tested |
| ☐ | ☒ | A description of any assumptions or corrections, such as tests of normality and adjustment for multiple comparisons |
| ☐ | ☒ | A full description of the statistical parameters including central tendency (e.g. means) or other basic estimates (e.g. regression coefficient) AND variation (e.g. standard deviation) or associated estimates of uncertainty (e.g. confidence intervals) |
| ☐ | ☒ | For null hypothesis testing, the test statistic (e.g. *F*, *t*, *r*) with confidence intervals, effect sizes, degrees of freedom and *P* value noted<br>*Give P values as exact values whenever suitable.* |
| ☒ | ☐ | For Bayesian analysis, information on the choice of priors and Markov chain Monte Carlo settings |
| ☒ | ☐ | For hierarchical and complex designs, identification of the appropriate level for tests and full reporting of outcomes |
| ☐ | ☒ | Estimates of effect sizes (e.g. Cohen's *d*, Pearson's *r*), indicating how they were calculated |

*Our web collection on statistics for biologists contains articles on many of the points above.*

## Software and code

Policy information about availability of computer code

| Data collection | No code used for data collection. |
|---|---|
| Data analysis | All routine analysis were performed using tools publicly available. DNBseq reads were filtered with SOAPnuke (version 2.1.7). Short/long reads were aligned using bowtie2 (2.2.8)/minimap2 (2.20-r1061). SNPs were called using tools shore (v 0.7.1). BAM, VCF file processing and sequencing depth analysis were performed using samtools (v1.9;v1.11) and bedtools (v2.29.1). PacBio sequence reads were assembled using hifiasm (v0.7). Hi-C data were processed using ALLHiC (v0.9.13), juicer (v1.6) and juicebox(v2.13.07). k-mers were processed with Jellyfish (version 2.2.10) and Merqury (version 1.3). Genome annotation was performed using EDTA (version 2.1.0),Repeatmasker (version 4.1.2-p1), BRAKER (version 2.1.6), HISAT2 (version 2.2.1), TSEBRA (version 1.0.3), liftoff (version 1.6.2), AGAT (version 1.0.0), and evaluated using BUSCO (version 5.2.2) .TRASH (version 1.2) was used for tandem repeat identifiction. Sequence alignment was performed using nucmer3 (version 3.1). Structural variations were identified using SyRI (v1.6) based on nucmer3 sequence alignments.rDNA, tDNA was detected using tRNAscan-SE (version 2.0.5), barrnap (version 0.9), and infernal (version 1.1.4) . The script infernal-tblout2gff.pl (version 1.1.4) was used to convert the result of infernal into GFF format. Sequence-level pan-genome was constructed using minigraph (version0.20-r55966); gene-level pan-genome was constructed usingOrthoFinder (version 2.5.5), diamond (version 2.0.13), and Blast (version 2.12.0+). mosdepth (version 0.3.1) was used for genomic coverage analysis. The distribution of SV along the genome was determined by identifying syntenic regions in all pairwise comparisons to DM using Msyd (version 1.0; https://github.com/schneebergerlab/msyd). Variant calling with long read alignment was performed using DeepVariant (v1.4.0)54. GVCF files were merged using GLnexus (v1.2.7). VCF files were split into windows using jvarkit (v2024.04.20) (https://github.com/lindenb/jvarkit). Maximum likelihood phylogenetic trees per window were constructed using IQ-TREE (v2.1.2). Consensus trees were built including all window trees per chromosome and for the whole genome using ASTRAL (v5.7.8). Genetic variants were pruned by LD using PLINK (1.90b6.18). Admixture analysis was conducted independently for each chromosome using ADMIXTURE (1.3.0). The package Dsuite (version 0.5) was used to calculate D-statistics, f4-ratio, and associated p-values in ABBA-BABA |

analysis. Customized code used in analysis supporting this work is available at https://github.com/HeQSun/tetraDecoder, https://github.com/schneeberglerlab/pantohap and https://github.com/schneeberglerlab/haplotype_threading.

For manuscripts utilizing custom algorithms or software that are central to the research but not yet described in published literature, software must be made available to editors and reviewers. We strongly encourage code deposition in a community repository (e.g. GitHub). See the Nature Portfolio guidelines for submitting code & software for further information.

## Data

Policy information about availability of data

All manuscripts must include a data availability statement. This statement should provide the following information, where applicable:
- Accession codes, unique identifiers, or web links for publicly available datasets
- A description of any restrictions on data availability
- For clinical datasets or third party data, please ensure that the statement adheres to our policy

The haplotype-resolved genome assemblies and the gene annotation files of the 11 varieties analyzed in this study are available at Zenodo (version 2: 10.5281/zenodo.10617012; version 2.1: 10.5281/zenodo.14053896). All high-throughput sequencing data generated in this study are available at NCBI under Bioproject PRJNA1074690. PacBio long read sequencing of 20 wild species were retrieved from NCBI under BioProject PRJNA754534, and PacBio long read, short read and Omni-C sequencings of cultivar 'Otava' were retrieved from NCBI under Bioproject PRJNA726019. Reference data (v6.1) of DM 1-3 516 R44 was downloaded at Spud DB (https://spuddb.uga.edu/download.shtml), and dAg1_v1.0 data was downloaded from figshare (doi.org/10.6084/m9.figshare.14604780).

## Research involving human participants, their data, or biological material

Policy information about studies with human participants or human data. See also policy information about sex, gender (identity/presentation), and sexual orientation and race, ethnicity and racism.

| | |
|---|---|
| Reporting on sex and gender | Not applicable. |
| Reporting on race, ethnicity, or other socially relevant groupings | Not applicable. |
| Population characteristics | Not applicable. |
| Recruitment | Not applicable. |
| Ethics oversight | Not applicable. |

Note that full information on the approval of the study protocol must also be provided in the manuscript.

# Field-specific reporting

Please select the one below that is the best fit for your research. If you are not sure, read the appropriate sections before making your selection.

☒ Life sciences ☐ Behavioural & social sciences ☐ Ecological, evolutionary & environmental sciences

For a reference copy of the document with all sections, see nature.com/documents/nr-reporting-summary-flat.pdf

# Life sciences study design

All studies must disclose on these points even when the disclosure is negative.

| | |
|---|---|
| Sample size | We searched the Wageningen University Potato Pedigree Database (with pedigree data from over 9,500 potato samples) for cultivars grown in the 19th century or those foundational to modern breeding efforts. From 164 cultivars in the database that fitted our criteria, 19 were available in the Gross Lüsewitz Potato Collections (GLKS) of the IPK Gene Bank. PCA analysis with whole-genome short-read sequencings of the 19 samples confirmed one sample as S. tuberosum spp. Andigena, and we selected the ten most diverse subset among the others (covering 90% of the variations among of the 19 samples) to generate the pan-genome of European potato (which was found to cover a high percentage (85%) of the haplotypes in tetraploid potatoes). |
| Data exclusions | By analyzing the variations using whole-genome short read sequencing among the nineteen varieties, we selected the ten representing over 90% of the variations of the nineteen varieties. Additionally, another cultivar 'Russet Burbank' was also assembled. |
| Replication | No biological or technical replicates were generated in this study, as it focused on the assembly and analysis of tetraploid potato genomes rather than experimental treatments.<br><br>However, to ensure the reproducibility and reliability of our findings, all genome assemblies were validated using multiple quality control measures, as detailed in the Supplementary Information. Assembly completeness was assessed using BUSCO, base accuracy was estimated through Merqury QV scores, and haplotype phasing accuracy was evaluated using Hi-C data. Additionally, our assembly methodology was evaluated by reconstructing a previously assembled genome, confirming the robustness of our approach. Sequence diversity analyses, haplotype reconstruction, and introgression detection were performed with clearly defined pipelines involving existing or newly developed |

bioinformatics tools, which are all publicly available. The assembled genomes and all relevant datasets generated or used in this study are also publicly available. All these will allow independent verification of our study.

Randomization | Randomization does not apply in  genome sequencing and assembly

Blinding | No group allocation was needed in this study.

# Reporting for specific materials, systems and methods

We require information from authors about some types of materials, experimental systems and methods used in many studies. Here, indicate whether each material, system or method listed is relevant to your study. If you are not sure if a list item applies to your research, read the appropriate section before selecting a response.

## Materials & experimental systems

| n/a | Involved in the study |
|---|---|
| ☒ ☐ | Antibodies |
| ☒ ☐ | Eukaryotic cell lines |
| ☒ ☐ | Palaeontology and archaeology |
| ☒ ☐ | Animals and other organisms |
| ☒ ☐ | Clinical data |
| ☒ ☐ | Dual use research of concern |
| ☐ ☒ | Plants |

## Methods

| n/a | Involved in the study |
|---|---|
| ☒ ☐ | ChIP-seq |
| ☒ ☐ | Flow cytometry |
| ☒ ☐ | MRI-based neuroimaging |

## Dual use research of concern

Policy information about dual use research of concern

### Hazards

Could the accidental, deliberate or reckless misuse of agents or technologies generated in the work, or the application of information presented in the manuscript, pose a threat to:

| No | Yes | |
|---|---|---|
| ☒ | ☐ | Public health |
| ☒ | ☐ | National security |
| ☒ | ☐ | Crops and/or livestock |
| ☒ | ☐ | Ecosystems |
| ☒ | ☐ | Any other significant area |

### Experiments of concern

Does the work involve any of these experiments of concern:

| No | Yes | |
|---|---|---|
| ☒ | ☐ | Demonstrate how to render a vaccine ineffective |
| ☒ | ☐ | Confer resistance to therapeutically useful antibiotics or antiviral agents |
| ☒ | ☐ | Enhance the virulence of a pathogen or render a nonpathogen virulent |
| ☒ | ☐ | Increase transmissibility of a pathogen |
| ☒ | ☐ | Alter the host range of a pathogen |
| ☒ | ☐ | Enable evasion of diagnostic/detection modalities |
| ☒ | ☐ | Enable the weaponization of a biological agent or toxin |
| ☒ | ☐ | Any other potentially harmful combination of experiments and agents |

## Plants

| | |
|---|---|
| Seed stocks | Materials were requested from the Gross Luesewitz Potato Collections (GLKS) of the IPK Gene Bank. |
| Novel plant genotypes | No novel palnt genenotype generated in this study. |
| Authentication | Not applicable. |

