## [Peer Review File · Nature]

The phased pan-genome of tetraploid European potato

Corresponding Author: Professor Korbinian Schneeberger

Version 0:

Reviewer comments:

Referee #1

(Remarks to the Author)

Sun et al. present in their manuscript the first haplotype-resolved chromosome-level pangenome of European potato landraces. This is a big sequencing and computational achievement that certainly will contribute to potato breeding and to enhancing a more informative use of resequenced genomes in the context of the now available European potato pangenome. The whole bioinformatic pipeline, from genome assembly to pangenome reconstructions is carefully done and validated. I find especially exciting and innovative the proposed use of the pan-genome graph for haplotype-phasing short-read data. However, the evolutionary genetics part of the manuscript is definitely not up to the standards of the pure bioinformatic analysis.

Major comments:

Their analysis of the patterns of selection is extremely superficial and relies only on two summary statistics of diversity (π and θ) and one summary statistic of the site frequency spectrum (Tajima's D). Reduction of diversity in conjunction with negative Tajima's D is not sufficient evidence to claim that a particular region of the genome has undergone positive selection as stated in lines 160 and 161, and especially in line 165, where the authors claimed that negative Tajima's D is an undoubtful sign of positive selection "local skews of negative Tajima's D values which indicate selective sweeps". This is not correct, as skews in the allele frequencies that yield negative values of Tajima's D could result from demographic processes such as population expansions following population bottlenecks. In line 171 they even refer to the regions with low diversity and high Tajima's D as "the selected regions" and in line 197 they assumed that a selected sweep indeed took place. Again, their analysis is superficial and their evidence for selection is extremely weak. Moreover, the authors did not consider the role of putative introgression or haplotype shuffling due to limited crossing on the levels of diversity and extent of linkage disequilibrium, nor did they correct for recombination rate when they used the levels of linkage disequilibrium as an additional indication of a putative selective sweep.

The introgression analysis is also preliminary and did not use any formal test of introgression. The authors did a good job identifying genomic blocks that are more similar to a *Solanum* wild species based on sequence similarity and read count. This analysis needs to be supplemented at least by a formal test of introgression such as D-statistics, which asks whether introgression took place considering the null hypothesis of incomplete lineage sorting. Moreover, the simplicity of the presented analysis is a missed opportunity to formally reconstruct the population history of modern European potatoes using an admixture graph approach. Furthermore, the use of UMAP, a rather controversial method to perform dimensionality reduction (distances between clusters are meaningless), to identify the likely sources of introgression is inadequate and does not provide any statistical support for the introgression sources, the proportion of the genome contributed by each source, or the timing of the introgressions.

The authors do not have any formal analysis regarding the timing of different introgressions, however, they point out in lines 211-214 that the identified introgressions are different from known breeding efforts to bring disease resistance that occurred during the second half of the XX century. In the same paragraph, they stated (Lines 214-215) "That implies that the introgressions in these founder cultivars might have been introduced during the adaptation to Europe or during domestication". This is a rather huge confidence interval for the timing of the introgression. Additionally, if *S. commersonnii* is their main candidate introgression donor, is there evidence of *S. commersonnii* being planted in Europe?

It will be also important to present a detailed analysis of the genes included in the putatively introgressed regions. For instance, are particularly gene families overrepresented in these regions? Are there any GO categories overrepresented?

Minor comments:

Line 65: The authors claim that “only potatoes preadapted to the new environment were successful in Europe” citing reference 12. This is incorrect, the evidence presented in reference 12 suggests that the adaptation of potatoes to European latitudes happened de novo in Europe.

Line 73: “the domestication in Latin America”. Incorrect, Latin America did not exist at the time of potato domestication, please refer to the region using only geographical terms: South America.

Figure 1B: Please explain how the ovals grouping PaB or the Group tuberosum were statistically constructed. Additionally, the zooming into the Group tuberosum is a bit misleading since the same percentages of variance explained are depicted in the x and y axes as in the PCA including Pb. Instead, the PCA can be recalculated using only potatoes belonging to the group tuberosum.

Line 248: Wrong reference: 13

(Remarks on code availability)

Referee #2

(Remarks to the Author)

Sun et al. present haplotype-resolved genome assemblies for 10 representative tetraploid European potato varieties, alongside a pan-genome graph constructed from these assembled haplotypes. The authors conducted comparative analyses of the haplotypes of these 10 varieties to identify domestication footprints and wild introgressions in the potato genome, and to characterize the haplotype diversity among European potatoes. Additionally, they attempted to demonstrate the utility of the constructed pan-genome graph in aiding the reconstruction of haplotypes for modern potatoes. While the haplotype-resolved genomes and pan-genome of European potatoes developed in this study provide valuable and unprecedented resources for potato research and breeding, my main concern lies in the predominantly descriptive nature of the results, lacking sufficient novel biological insights.

Major:

The authors assessed the phased assemblies with Merqury and BUSCO. Nonetheless, it's important to note that these assessments can only suggest the completeness and base accuracy of the assemblies. A major challenge in haplotype-resolved assemblies is phase switching. Despite the authors' evaluation of their assembly pipeline using their previously assembled 'Otava' genome, phase switching still needs to be assessed for the assembled genomes reported in this study. Although the authors provided Hi-C map for each assembled haplotype (Figs. S29-38), this type of presentation will not reveal phase switching errors, if there is any. Hi-C heatmaps showing each set of the four haplotypes from the same chromosomes should be presented together in one map, which would offer a clearer indication of the accuracy of the assemblies in terms of phasing.

Line 129-223: While these two sections provide some useful information on the domestication history and genetic diversity of European potatoes, they are largely descriptive, mainly presenting some statistics without deep biological insights.

Line 3267-307: In this section, the authors aimed to showcase the utility of the constructed pan-genome graph in reconstructing haplotypes of modern potatoes using Illumina short reads. However, only approximately 48% of the haplotypes could be reconstructed (Line 300). As the pan-genome graph has already encompassed the majority of the diversity (Line 224), incorporating additional haplotypes to the graph would not substantially increase this value. Consequently, the proposed haplotype reconstruction approach appears to offer limited practical value.

Minor:

Line 113: Reference 1 was wrongly cited here.

Line 144, “yellow flesh is to a single haplotype”: grammar error.

Line 158-160: It's hard for me to understand this sentence. How chromosome 10 can encode differences of tuber shapes and skin and flesh colors?

Figs. S3-14 are not readable. The resolution is too low, and the font is too small. Same for some other figures. Please check.

Fig. S22: The x-axis is not clear. Log10 of what?

(Remarks on code availability)

Referee #3

(Remarks to the Author)

In their manuscript, Sun and colleagues describe genome assembly and comparative analysis of key founder lines of European potato and construction of a pan-genome resource for this breeding population. The authors demonstrate that high nucleotide diversity has been retained in this population, despite multiple genetic bottlenecks in its history due to the persistence of a small number of very diverse haplotypes. The authors further characterize the set of core and dispensable genes within potato genomes and provide evidence of introgression of wild potato alleles into the European potato population. Finally, the authors demonstrate the utility of their pan-genome resource through reconstruction of haplotypes using short-read data from one of their assembled genomes. The authors have successfully leveraged recent technological innovations in sequencing to assemble a complex tetraploid genome and have created a resource that should be valuable for the European potato breeding community. I believe the study is well executed, but I have recommendations the authors may wish to consider for improving the manuscript and I have concerns about the manuscript meeting criteria of novelty and significance typically required for publication in Nature.

1) The result showing high nucleotide diversity and a small number of haplotypes is very believable, but seems fairly intuitive given the small number of generations of breeding that have occurred in European potato. One thing the authors do not mention directly is that potato is often vegetatively propagated, which limits the extent of meiosis and recombination. Low recombination regions near centromeres or within inversions may show even more extreme blocks of linkage, but this is not a hypothesis that the authors explicitly test. Simulations showing expected haplotype lengths given the approximate number of generations of sexual reproduction and analyses testing enrichment of long haplotypes in inversions and pericentromeric regions would be helpful for gauging the extent to which these empirical results are expected given what we know about the history and genetic architecture of European potato.

2) It was unclear from the supplementary methods exactly how gene models were lifted over between contigs. Was a single genome copy annotated and then these annotations were lifted over to the other haploid chromosomes or were all copies annotated and lift over was done amongst all copies to fill in annotation gaps? The former approach would seem to miss a fair number of genes present within a particular genome as PAV across homologous chromosomes.

3) The low-level of haplotype assignment (48%) in the test case for the pan-genome seems concerning given the intention for this to serve as resource for haplotype construction.

4) Construction of crop pan-genomes is a very active area of research with several graph pan-genomes published over the last few years. A few examples include:

<https://www.sciencedirect.com/science/article/pii/S2590346223003498>

<https://genomebiology.biomedcentral.com/articles/10.1186/s13059-023-03133-2>

<https://www.nature.com/articles/s41422-022-00685-z>

Many of these studies are more comprehensive in terms of global diversity of their respective crops than the current study; the pan-genome described here will likely serve as an effective resource for European potato only.

Furthermore, I do not see innovations in the approaches included here that push the field forward or that would be widely adopted across systems.

In summary, I think this is a well-executed study, I believe the authors have generated resources that will be very useful for the European potato breeding community, and I believe the manuscript is well-written and the results are clear. The study would find a good publication home in a more disciplinary journal.

(Remarks on code availability)

Analyses and computational pipelines are well-documented in the accompanying repository.

Version 1:

Reviewer comments:

Referee #1

(Remarks to the Author)

I thank Sun et al. for their response to my comments, as well as those of the other reviewers. I believe that the incorporation of the reviewers' suggestions has improved the quality of the manuscript. The bioinformatics section of the manuscript is of high quality, and the new validations of the haplotype-graph method are truly impressive. However, the evolutionary genetics section of the manuscript is still unsatisfactory, and the authors' responses fall short of addressing the concerns or are not correctly interpreted. I will describe below the main points of my critique and my concrete suggestions to get this manuscript published:

-Positive selection

My original critique stated that reduction of diversity in conjunction with negative Tajima's D is not sufficient evidence to claim that a particular region of the genome has undergone positive selection and that a negative Tajima's D does not necessarily imply positive selection.

To reply to my critique the authors investigated the relationship between genetic diversity and recombination rate, as well as genetic diversity and LD, observing a significant positive and negative R^2 , respectively (Fig. S29A-B). The positive correlation between genetic diversity and recombination rate has been recurrently observed among different taxa (reviewed in Kern and Hahn, MBE, 2018; but see also Corbett-Detig, PLoS Biology, 2015). This positive correlation is interpreted as the effect of linked selection on polymorphism. Linked selection can be the result of positive selection (hitchhiking) and/or purifying selection (background selection). Disentangling between these two alternatives across the genome is extremely difficult, but linked selection needs to be invoked to explain the patterns of Fig. S29A-B. Linked selection was not explicitly mentioned by the authors in their response or incorporated in the manuscript.

In contrast to what the authors claim: "To test the effects of demography on genomic diversity, we correlated LD values with genetic diversity as well as with local recombination rate along the chromosomes", they have not tested the effects of demography at all. The above mentioned positive correlation between genetic diversity and recombination rate will be present with or without the presence of bottlenecks. Naturally, levels of LD are influenced by demography—population contractions increase LD, while expansions decrease it. Therefore, it stands to reason that repeated bottlenecks have raised LD levels, making it more difficult to detect positive selection. Formally testing the influence of demography would require explicitly modeling demographic scenarios and comparing them against your data. However, I am not suggesting that the authors take this approach.

Finally, the mixed model used by the authors to quantify the contributions of local recombination rate, the presence of structural variants, and chromosome identity to LD is not sufficient to support a claim of positive selection. The LD that is not explained by these factors could be in part caused by linked selection (positive and purifying selection in unknown proportions). The length of the putatively selected region on Chr10, along with its pericentromeric location, makes it unlikely that positive selection is the underlying cause of the observed pattern, since the selection coefficient that would be required to produce a sweep signal of this magnitude likely needs to be very high.

In the rebuttal, the authors refer to purifying selection as the reason for the reduced level of genetic diversity and lack of introgression in the region of Chr10. This is the only instance, where the authors mention purifying selection, which is not mentioned in the manuscript at all. It is confusing since in all other instances the authors seem to be referring to positive selection, when they say selection. As I mentioned above, I believe purifying selection is responsible for some unknown proportion of the pattern.

My suggestion is to remove this entire section from the manuscript. Rather than adding value to the excellent bioinformatics work, it undermines it. The absence of any reference to this section in either the abstract or the discussion further supports my suggestion to remove it from the manuscript.

-Introgression

I appreciate that the authors present a quantitative analysis of introgression using D-statistics and f₄-statistics. However, the analyses are not reported using current standards and the significance is not assessed in the proper way. D-statistics and f₄-statistics are meant to identify genome-wide patterns left by introgression. The significance of the test is evaluated by a block-jackknife procedure, which is meant to correct for the extent of LD in the assessed individuals/populations.

The authors should clearly present the tree configuration used in the D-statistics as (A,B;C,D), in which D is the outgroup, C is the candidate for introgression and A and B are the evaluated individual/populations. It is particularly important to report the outgroup used in the analysis, since the test requires the outgroup to be far enough and equally distant from the tested populations. The analysis is difficult to follow due to the non-standard way in which the analyses are reported.

You should only report genome-wide tests and specify the size of the jackknife blocks used in the analysis. The blocksize should be longer than the genome-wide LD decay. Window-based D-statistics should be avoided, as there is now way to test for statistical significance for each window, as the jackknife test of significance cannot be performed. None of the other D-statistic-inspired tests are adequate to be performed on windows.

The interpretation of the Admixture analysis presented in Figure 3b is not adequate. The authors stated that "genome-wide admixture analyses revealed clear signatures of admixture specific to the European cultivars." Admixture/Structure plots cannot be used to infer a history of admixture/introgression, despite their suggestive names. The software only tries to minimize LD and Hardy-Weinberg disequilibrium for a given number of Ks and infer ancestry components. Multiple scenarios not involving introgression can give rise to patterns of mixed ancestry in populations, i.e. inclusion of diverse populations and subpopulations that underwent population bottlenecks.

Please indicate in the main text the optimal K that was identified.

In summary, the introgression section needs extensive reanalysis and needs to be rewritten.

Other comments:

Figure 3

Figure 3A: Clearly indicate in the phylogeny which species belong to clade C4N and C4S.

Figure 3D-E can be removed, as the per-window based tests are not recommended (as explained above). You can replace this by the equivalent but only for genome-wide tests as described above (reporting the tree configuration, significance and jackknife block size).

(Remarks on code availability)

Referee #2

(Remarks to the Author)

This revised version has been substantially improved. I only have a few minor comments remaining.

Haplotype sharing: The details on how shared haplotypes are defined seem to be missing. Are two haplotype blocks with identical sequences considered shared? Please clarify.

The authors predicted between 36,862 and 47,316 protein-coding genes in the 40 haploid genomes. This large variation in the number of predicted genes should be explained or discussed.

Page 4: "multiple thousands contigs" -> "multiple thousands of contigs".

"generate a recombinant population for each of the tetraploid genomes": change "genomes" to "cultivars"

Page 5, "Even though.....tuber flesh": this is not a complete sentence.

Page 7: The term "chromosome identity" needs to be defined more clearly. What exactly does it refer to in this context?

Page 23, legend of Fig. 3b: Please specify which chromosome's result is shown here.

(Remarks on code availability)

Version 2:

Reviewer comments:

Referee #1

(Remarks to the Author)

I thank Sun et al. for their response to my comments and for presenting a more concise manuscript.

I comment here on their response to my comments on the selection and introgression section. Quoted text refers to Sun et al. response to my comment or their to new version of the manuscript.

-Selection

The selection section is more concise, with its interpretation more nuanced, acknowledging that their approach cannot disentangle the linked effects of positive and purifying selection. Their claim of positive selection is now appropriately framed as speculative: "Even though linked selection includes the combined effects of purifying and positive selection, we speculated that the latter played an important role in chromosome 10".

I still believe this section adds very little to the manuscript, as detecting the footprints of linked selection is unsurprising. Moreover, their approach fails to distinguish between the linked effects of positive and purifying selection or to identify candidates of positive selection.

In their response the authors state that "We would like to point out that this does not affect our main conclusions (that selection acted on the potato genomes during domestication and/or the transition to Europe)". This statement is very likely to be true—selection is constantly acting on genomes. I also agree that positive selection likely played a role during domestication and the transition to Europe. However, the challenge lies in detecting the footprints of positive selection and identifying the loci it targeted. The authors fell short of achieving these objectives.

As I mentioned in my previous comments, the authors still do not make any reference to the selection section in the abstract,

further supporting the little relevance of this part for the overall study. I leave the decision regarding the inclusion of this section to the authors and the editor.

-Introgression

The reporting D-statistics and F4 statistics used in the revised manuscript now follows the standards of the field. The authors used two different configurations for the D-statistics:

D(C4S, C4S; Cultivar, C3) (they called this configuration C4S).

D(Cultivar, C4S; C3, C1+2) (they called this configuration C3).

Configuration I. follows the tree topology presented in figure 3a. This D-statistics uses C3 as an outgroup, thus the test assumes that C3 cannot take part in any introgression and always represents the ancestral state. In contrast, configuration II. uses C3 as a possible contributor to the introgression. Both three topologies I. and II. cannot be true at the same time. In my opinion topology I. is more appropriate, and if this is the case, topology II. violates the assumptions of the test and is an invalid use of the test.

I suggest removing configuration II. Moreover, a more standard way to organize the configuration would be D(Cultivar, Cultivar; C4S, C3), iterating along all possible samples of Cultivars and C4S and keeping C3 fixed.

The authors stated in their reply that “The display of D-statistics at window level is not a new approach and has been employed before”. Indeed, it is not new but it is incorrect. The expectation of $D=0$ if no introgression took place is valid only when averaged across many loci throughout the genome. As I explained in my last comments, it is not possible to test for the statistical significance of any identified region. D-statistics different from zero in genomic-windows can indicate local sorting of the discordant topologies used in the test and not necessarily introgression. Therefore, the test was developed as a genome-wide statistic, however the use of D-statistic at a chromosome scale is valid.

(Remarks on code availability)

Referee #2

(Remarks to the Author)

The authors have addressed all my concerns. I congratulate the authors for a very interesting study!

(Remarks on code availability)

Referee #1

Sun et al. present in their manuscript the first haplotype-resolved chromosome-level pangenome of European potato landraces. This is a big sequencing and computational achievement that certainly will contribute to potato breeding and to enhancing a more informative use of resequenced genomes in the context of the now available European potato pangenome. The whole bioinformatic pipeline, from genome assembly to pangenome reconstructions is carefully done and validated. I find especially exciting and innovative the proposed use of the pan-genome graph for haplotype-phasing short-read data. However, the evolutionary genetics part of the manuscript is definitely not up to the standards of the pure bioinformatic analysis.

Thank you very much for pointing out the relevance of our project and the careful work that we put into the assembly of the genomes. Guided by your comments, we have advanced the evolutionary genetics analysis of the ten potato samples, which obviously improved our search for footprints of selection.

Major comments:

1. Their analysis of the patterns of selection is extremely superficial and relies only on two summary statistics of diversity (π and θ) and one summary statistic of the site frequency spectrum (Tajima's D). Reduction of diversity in conjunction with negative Tajima's D is not sufficient evidence to claim that a particular region of the genome has undergone positive selection as stated in lines 160 and 161, and especially in line 165, where the authors claimed that negative Tajima's D is an undoubtful sign of positive selection "local skews of negative Tajima's D values which indicate selective sweeps". This is not correct, as skews in the allele frequencies that yield negative values of Tajima's D could result from demographic processes such as population expansions following population bottlenecks. In line 171 they even refer to the regions with low diversity and high Tajima's D as "the selected regions" and in line 197 they assumed that a selected sweep indeed took place. Again, their analysis is superficial and their evidence for selection is extremely weak.

We appreciate your detailed feedback. We agree that negative Tajima's D values can result from demographic processes as well, and that low Tajima's D alone is not sufficient for the identification of patterns of selection. We have revised our manuscript to address this point comprehensively and thereby refined our conclusions on selected regions, which now zooms in on one region on chromosome 10, which still displays itself as outlier.

Our initial analysis identified regions with low diversity and negative Tajima's D as potential candidates for positive selection. While these patterns can also arise from demographic changes such as population bottlenecks and expansions, we would

expect demographic effects to impact the entire genome, rather than being confined to specific local regions. However, as suggested, the effect of recombination can amplify these demographic effects locally, in particular in regions with low recombination where linkage disequilibrium (LD) is high. To test if the regions with low diversity are influenced by this factor, we evaluated the correlation of genetic diversity and LD along the genome, including the variation of meiotic recombination. And indeed, this was the case. A new analysis revealed a negative correlation between LD and genetic diversity, as well as between LD and recombination rates, which is consistent with the effects of demographic changes.

However, despite this genome-wide trend, we speculated that recombination alone would not fully explain the extreme values that we observed in the candidate regions. Using a mixed model that accounts for local recombination rate, structural variants, and chromosomal effects on LD, we calculated the residuals of the observed values to the model for each genomic position (see also our answer to your comment #2). If no other constraints would affect the genome, we would expect that the residuals of the mixed model analysis are randomly distributed across the genome. However, the top 5% of most extreme residuals were almost exclusively found in one of the previously identified candidate regions on chromosome 10 (new Fig. 4).

Additionally, we now show that the regional reduction in diversity, in particular the reduction in chromosome 10, cannot be observed in wild species (Supplementary Fig 30). This makes the reduction in diversity specific to the cultivars, and further supports that at least the region on chromosome 10 is under selection, and that the extreme values are not due to demographic effects alone.

Furthermore, we also extended our introgression analyses (as per your comment #2) and observed significant variation in introgression between haplotypes. Notably, however, the region on chromosome 10 was homogeneous and showed a marked depletion of introgressions compared to other chromosomes or regions (new Fig. 3g). The lack of introgression and variation in this candidate region further supports the hypothesis of consistent purifying selection acting on these haplotypes.

Taken together, based on a careful analysis of the candidate regions to test the effect of demography, we found that one of the candidate regions shows signatures of selection, which is in line with the fact that some of the alleles that code for the marked differences in phenotypes between European and South American potatoes are in fact located on chromosome 10.

2. Moreover, the authors did not consider the role of putative introgression or haplotype shuffling due to limited crossing on the levels of diversity and extent of linkage disequilibrium, nor did they correct for recombination rate when they used the levels of linkage disequilibrium as an additional indication of a putative selective sweep.

As mentioned, we have now incorporated recombination rate information to correct for its influence on linkage disequilibrium (LD). Additionally, we have expanded our analysis to consider the role of introgression and haplotype shuffling. For this, we performed an assessment of introgressions across the genome and the potential source of introgression (clade) (Fig. 3). We use this information to calculate the number of introgressed haplotypes and diversity along the genome. This information was included in the mix model to account for ancestry.

While it is most likely true that only a few sexual generations have occurred in Europe (arrival in Europe less than 500 years ago, and clonal propagation), it is crucial to consider that the observed introgressions most likely occurred before the transport to Europe, probably during and/or after the domestication process in South America (~10,000 years ago). However, the exact number of sexual generations since domestication started is not known.

3. The introgression analysis is also preliminary and did not use any formal test of introgression. The authors did a good job identifying genomic blocks that are more similar to a *Solanum* wild species based on sequence similarity and read count. This analysis needs to be supplemented at least by a formal test of introgression such as D-statistics, which asks whether introgression took place considering the null hypothesis of incomplete lineage sorting.

Again, we appreciate your comment very much as it strongly improved this part of our study. In our initial analysis on introgression, we followed a recent study, where introgressions in a potato cultivar were identified via read alignments (Bao et al, *Molecular Plant* 2022). We acknowledge that this approach, while useful, was rather simplistic. However, it was supported by the fact that there is high divergence between wild species involved, making incomplete lineage sorting an unlikely explanation for the observed sequence similarities.

That said, we agree with the reviewer that this should be supplemented by formal testing and we changed this part of the analysis to a more formal approach. We now describe the results of D-statistics (ABBA-BABA) and f_4 statistics (new Fig 3). Our new analyses revealed strong signals of introgression and gene flow from wild species across the entire genome. Specifically, we observed significant introgressions from species within the C4S clade, primarily composed of wild species from southern South America, which is geographically distant from the traditional domestication centers in the Andes.

4. Moreover, the simplicity of the presented analysis is a missed opportunity to formally reconstruct the population history of modern European potatoes using an admixture graph approach.

We agree that the reconstruction of the population history of European potatoes is an intriguing topic, but we think this is beyond the possibilities of this study. Our sampling strategy was specifically designed to *maximize genetic diversity* and to capture the *ancestral* genetic variation rather than to represent the current population structure of European potatoes. Consequently, our sample set is not adequate for a comprehensive reconstruction of the population history of modern European potatoes. Instead, our approach was aimed at providing insights into the genetic foundation and diversity used for breeding purposes rather than understanding the detailed population dynamics of contemporary European potatoes.

5. Furthermore, the use of UMAP, a rather controversial method to perform dimensionality reduction (distances between clusters are meaningless), to identify the likely sources of introgression is inadequate and does not provide any statistical support for the introgression sources, the proportion of the genome contributed by each source, or the timing of the introgressions.

We agree with your concerns regarding the use of UMAP for identifying introgressions. We have completely removed the UMAP analysis from our study. Instead, we have replaced it with a formal introgression analysis as described above.

6. The authors do not have any formal analysis regarding the timing of different introgressions, however, they point out in lines 211-214 that the identified introgressions are different from known breeding efforts to bring disease resistance that occurred during the second half of the XX century. In the same paragraph, they stated (Lines 214-215) "That implies that the introgressions in these founder cultivars might have been introduced during the adaptation to Europe or during domestication". This is a rather huge confidence interval for the timing of the introgression.

The samples that we selected for our pan-genome analysis were introduced before introgression breeding within the European breeding programs began (which is well recorded). Therefore, the introgressions in these genomes must have occurred outside of modern breeding efforts. Instead of precise introgression timing, our aim is to state that there are many introgressions in European potatoes (including some of multiple tens of Mb in length) that are *not* the result of modern breeding efforts as this is the prevailing opinion on extended introgressions. In the new version of the manuscript we make this point clear.

7. Additionally, if *S. commersonnii* is their main candidate introgression donor, is there evidence of *S. commersonnii* being planted in Europe?

It is true that *S. commersonnii* is not endemic to Europe, making it unlikely to be the source of spontaneous introgression into potato in Europe, even though such introgressions could have happened before the transition to Europe (please also see our answer to comment #6). Even more, *S. commersonnii* and *S. tuberosum* are sexually incompatible and require a bridging species for successful hybridization. Therefore, it is also plausible that a related, compatible species, such as *S. okadae*, served as the actual donor of the introgressions that were identified via sequence similarity to *S. commersonnii*.

Generally, it is important to note that while sequence similarity may suggest a particular species or clade, it does not identify the actual source species of the introgression. The observed introgressions could originate from related species within the same clade. To make this more clear, in our revised version, we describe the clades of potential introgression sources rather than pinpointing specific species.

8. It will be also important to present a detailed analysis of the genes included in the putatively introgressed regions. For instance, are particularly gene families overrepresented in these regions? Are there any GO categories overrepresented?

Selection for introgressed regions is usually based on a limited number of alleles within the introgressed regions. The introgressed regions therefore include many more genes which are not related to the actual reason for the maintenance of the introgression (*e.g.*, introgressions for resistances typically only introgress one functional allele while such introgressions can be megabase-scale in size) and thus, GO analysis of introgressed regions are unlikely to give insights into the reason for the maintenance of the introgressions.

Minor comments:

1. Line 65: The authors claim that “only potatoes preadapted to the new environment were successful in Europe” citing reference 12. This is incorrect, the evidence presented in reference 12 suggests that the adaptation of potatoes to European latitudes happened *de novo* in Europe.

Thank you for the clarification of our misunderstanding. We have revised the text.

2. Line 73: “the domestication in Latin America”. Incorrect, Latin America did not exist at the time of potato domestication, please refer to the region using only geographical terms: South America.

We agree, thank you for this comment. We changed it at all occurrences.

3. Figure 1B: Please explain how the ovals grouping PaB or the Group tuberosum were statistically constructed. Additionally, the zooming into the Group tuberosum is a bit misleading since the same percentages of variance explained are depicted in the x and y axes as in the PCA including Pb. Instead, the PCA can be recalculated using only potatoes belonging to the group tuberosum.

The clusters were not statistically constructed but based on the clear separation of the one sample as compared to all others on PC1. In fact, the one sample that creates the first cluster turned out to be a potato sample of a different subspecies as compared to all other samples. This subspecies never settled in mainland Europe (the respective sample is from the Canary Island) and was not used for breeding of the European elite cultivars. However, as it matched our initial search criteria for clones that were present in Europe during the times breeding started, we had also sequenced it. To avoid misunderstandings, we have removed the zoom-in plot entirely.

4. Line 248: Wrong reference: 13

Thank you, we replaced it.

Referee #2 (Remarks to the Author):

Sun et al. present haplotype-resolved genome assemblies for 10 representative tetraploid European potato varieties, alongside a pan-genome graph constructed from these assembled haplotypes. The authors conducted comparative analyses of the haplotypes of these 10 varieties to identify domestication footprints and wild introgressions in the potato genome, and to characterize the haplotype diversity among European potatoes. Additionally, they attempted to demonstrate the utility of the constructed pan-genome graph in aiding the reconstruction of haplotypes for modern potatoes. While the haplotype-resolved genomes and pan-genome of European potatoes developed in this study provide valuable and unprecedented resources for potato research and breeding, my main concern lies in the predominantly descriptive nature of the results, lacking sufficient novel biological insights.

Thank you very much for your efforts to review our manuscript. We appreciate your notion that the new assemblies are a valuable and unprecedented resource for potato research and breeding.

Major comments:

1. The authors assessed the phased assemblies with Merqury and BUSCO. Nonetheless, it's important to note that these assessments can only suggest the completeness and base accuracy of the assemblies. A major challenge in haplotype-resolved assemblies is phase switching. Despite the authors' evaluation of their assembly pipeline using their previously assembled 'Otava' genome, phase switching still needs to be assessed for the assembled genomes reported in this study. Although the authors provided Hi-C map for each assembled haplotype (Figs. S29-38), this type of presentation will not reveal phase switching errors, if there is any. Hi-C heatmaps showing each set of the four haplotypes from the same chromosomes should be presented together in one map, which would offer a clearer indication of the accuracy of the assemblies in terms of phasing.

Thank you very much for this valuable suggestion. The analysis of Hi-C heatmaps with all four haplotypes triggered us to not only evaluate haplotype switch errors but also to further improve the assemblies. First, we generated Hi-C heatmaps including all four haplotypes of the same chromosomes and analyzed them for haplotype switch errors. Overall, we found and corrected 24 such errors within the 120 chromosomes of the ten cultivars. An example of the identification and correction of a haplotype switch error is shown below (Supplementary Figure r1).

All Hi-C heatmaps of the four haplotypes of each chromosome of the new assemblies have been provided as Supplementary figures 36-45 (as well as in Supplementary figure 46, where we describe the newly added assembly of 'Russet Burbank').

Supplementary Figure r1. Identification and correction of haplotype switching errors using Hi-C heatmaps across the four haplotypes of the same chromosome (here chromosome 7 of ‘Flava’). The left panel shows the Hi-C heatmap (haplotypes indicated in cyan) before correction, while the right panel shows that the heatmap after correction. In the left panel, the Hi-C heatmap of haplotype 3 was divided into three major blocks a, X and b (or symmetrically a', X', b') following the clear borders between them (*i.e.*, the non-smooth changes in contact intensity between these blocks that reveal haplotype switching errors). These patterns imply that the sequence of haplotype 3 should be divided into three regions r31, r32 and r33. Similarly, also haplotype 4 shows similar patterns, which suggest that its sequence should be divided into three segmental regions r41, r42, and r43. The strong contact intensity in block c/c' implied that region r31 in haplotype 3 was in strong contact with region r42 in haplotype 4, while block e/e' implied that region r41 in haplotype 4 was in strong contact with region r32 in haplotype 3. Similarly, the contact intensity in block f/f' suggested that region r33 in haplotype 3 was in strong contact with region r42 in haplotype 4, while the patterns in block d/d' implied that region r43 in haplotype 4 was in strong contact with region r32 in haplotype 3. Together, these patterns suggested that r32 in current haplotype 3 (bar in red) and r42 in current haplotype 4 (bar in dark blue) need to be switched and that the current arrangement is a haplotype switching error. After correction, as shown in the right panel, a new haplotype 3' was formed by integrating r31-r42-r33 and a new haplotype 4' was formed by integrating r41-r32-43. The contact intensity patterns were rearranged correspondingly leading to more smooth change in contact intensity within the new haplotypes as compared to the original version.

2. Line 129-223: While these two sections provide some useful information on the domestication history and genetic diversity of European potatoes, they are largely descriptive, mainly presenting some statistics without deep biological insights.

In response to this comment and to several comments by reviewer #1, we have undertaken extensive revisions of this section. The advanced analyses have led to several important insights that were not previously recognized:

Reduced Haplotype Diversity: Contrary to the prevailing belief that potato populations exhibit very high diversity, our study reveals a strongly reduced haplotype diversity. This has important implications in plant breeding and a new perspective on the genetic variability in cultivated potatoes. Further, the identification of the reduced haplotype diversity was prerequisite to our efforts to generate a pan-genome graph for the reconstruction of additional potato genomes.

Divergent Haplotype Ancestry: We show significant divergence between haplotypes, indicating diverse ancestry.

Extensive Introgression: Through formal introgression analyses, including D-statistics (ABBA-BABA) and f_4 statistics, we demonstrated extensive introgression from multiple divergent groups. Importantly, we established that these introgressions are not a result of recent breeding efforts as generally thought but likely already occurred during domestication.

Regions Under Selection: Using a mixed model that accounts for local recombination rate, structural variants, and chromosomal effects, we identified candidate regions under selection, showing strong local reduction in diversity, negative Tajima's D and an increase in linkage disequilibrium. This selection is potentially involved in domestication since the same regions did not show reduction in diversity for wild species. Concurrently, we observed generalized extended linkage disequilibrium, which has important implications for past and future association studies on domestication traits.

3. Line 3267-307: In this section, the authors aimed to showcase the utility of the constructed pan-genome graph in reconstructing haplotypes of modern potatoes using Illumina short reads. However, only approximately 48% of the haplotypes could be reconstructed (Line 300). As the pan-genome graph has already encompassed the majority of the diversity (Line 224), incorporating additional haplotypes to the graph would not substantially increase this value. Consequently, the proposed haplotype reconstruction approach appears to offer limited practical value.

The first version of the graph-based genome reconstruction did not attempt the reconstruction of the entire genome, but was focused on regions that were co-linear

across all haplotypes. While this was an enormous improvement over conventional genotyping methods (e.g., like the commonly used Infinium 8k SNP Array) we agree that it is not comparable to the results of an actual genome assembly.

During the revision of this manuscript, we have extensively improved our genome-graph method. The updated method is now based on *all* haplotype-specific *k*-mers in *all* regions of the genome. With this we can reconstruct ~80% of the haplotypes of the genome of a modern potato cultivar using short reads only. Considering the repetitiveness of potato genomes and the size of the pericentromeric regions, we would like to emphasize that reconstructing 80% of the individual haplotypes of a tetraploid genomes with short reads is unprecedented.

Furthermore, we now demonstrate the usage of the graph in three different scenarios. In the first case, we reconstructed the genome of 'White Rose', a cultivar that was used to build the graph. In this optimistic use-case, we reconstructed haplotypes on a Mb scale implying that it is possible to reconstruct haplotype-resolved genome sequences with short reads if they are included in the graph. In the second case, we reconstruct the genome of 'Kenva', a cultivar that was derived from three cultivars that are included in the graph. We therefore expect that all haplotypes of 'Kenva' are in the graph, but that they are recombined. This simulated a realistic future use case, once we can use this method with a complete pan-genome. Again, the reconstruction of the Kenva genome led to Mb-scale haplotype-resolved regions.

The third use case is the most realistic use case. Here we used the genome-graph to reconstruct the genome sequence of 'Russet Burbank' with short reads only. 'Russet Burbank' is used for ~70% of the global French Fries production, but despite the importance of this cultivar no public genome assembly is available so far.

The reconstruction led to a genome assembly of 2.03 Gb with an N50 of 0.60 Mb. To validate the assembly, we have, in addition, generated a *de novo* genome assembly of 'Russet Burbank' (following the same procedure as we did for the 10 other cultivars, which led to a chromosome-level, haplotype-resolved assembly). We aligned the contigs from the genome-graph reconstruction to the *de novo* assembly and found that almost 90% of the contigs fully aligned to individual haplotypes of the *de novo* assembly, including an impressive, 9.9 Mb long contig (Supplementary Figure 31).

Taken together, the successful genome reconstruction of modern potato cultivars using the haplotype-graph outlined that the pan-genome will serve as a cost-effective method to analyze the genomes of large panels of potatoes.

Minor comments:

1. Line 113: Reference 1 was wrongly cited here.

Thanks, we have removed the reference.

2. Line 144, “yellow flesh is to a single haplotype”: grammar error.

This sentence was removed entirely.

3. Line 158-160: It’s hard for me to understand this sentence. How chromosome 10 can encode differences of tuber shapes and skin and flesh colors?

Multiple genes which encode alleles that cause tuber differences have been mapped to chromosome 10. We have revised the sentence.

4. Figs. S3-14 are not readable. The resolution is too low, and the font is too small. Same for some other figures. Please check.

Thank you for this helpful comment. We have updated the figures with a higher resolution.

5. Fig. S22: The x-axis is not clear. Log₁₀ of what?

Agreed. We have edited the Supplementary figure. It corresponds to the log₁₀ of the genetic distance in base pairs.

Referee #3 (Remarks to the Author):

In their manuscript, Sun and colleagues describe genome assembly and comparative analysis of key founder lines of European potato and construction of a pan-genome resource for this breeding population. The authors demonstrate that high nucleotide diversity has been retained in this population, despite multiple genetic bottlenecks in its history due to the persistence of a small number of very diverse haplotypes. The authors further characterize the set of core and dispensable genes within potato genomes and provide evidence of introgression of wild potato alleles into the European potato population. Finally, the authors demonstrate the utility of their pan-genome resource through reconstruction of haplotypes using short-read data from one of their assembled genomes. The authors have successfully leveraged recent technological innovations in sequencing to assemble a complex tetraploid genome and have created a resource that should be valuable for the European potato breeding community. I believe the study is well executed, but I have recommendations the authors may wish to consider for improving the manuscript and I have concerns about the manuscript meeting criteria of novelty and significance typically required for publication in Nature.

Thank you very much for taking the time to review our manuscript and pointing out that it was well executed.

The result showing high nucleotide diversity and a small number of haplotypes is very believable, but seems fairly intuitive given the small number of generations of breeding that have occurred in European potato. One thing the authors do not mention directly is that potato is often vegetatively propagated, which limits the extent of meiosis and recombination. Low recombination regions near centromeres or within inversions may show even more extreme blocks of linkage, but this is not a hypothesis that the authors explicitly test.

In our revised manuscript, we incorporated actual recombination rate data in the analyses of LD and low diversity during our search for patterns of selection (please also see our response #1 to reviewer 1). As you suggested, we indeed observed a negative correlation between LD and recombination rates, as well as between LD and genetic diversity. This is consistent with the effects of demographic changes, yet does not fully explain the extreme LD, reduced genetic diversity, or negative Tajima's D values in some candidate regions. We therefore used a mixed model accounting for local recombination rate, structural variants (SV), and chromosomal effects, and we identified that the regions with the top 5% of extreme residuals primarily map to one of our candidate regions, suggesting the pattern of low diversity in this region are not due to demographic effects or structural variation alone.

Furthermore, we would like to clarify that, although there is probably only a small number of generations that have occurred in Europe, there is no direct evidence to support the same during or after domestication. The number of sexual generations since the beginning of domestication is not obvious. Our sampling was primarily focused on the parental population of the Europe breeding programmes. Consequently, the ancestry blocks observed in our study reflect genetic events that occurred during or after domestication, rather than being influenced by the limited number of crosses in Europe after modern breeding began.

Simulations showing expected haplotype lengths given the approximate number of generations of sexual reproduction and analyses testing enrichment of long haplotypes in inversions and pericentromeric regions would be helpful for gauging the extent to which these empirical results are expected given what we know about the history and genetic architecture of European potato.

We fully agree that the reconstruction of the breakdown of ancestral haplotypes over sexual generations is an interesting question. However, there are two major limitations in the sampling of the ten potatoes that would hinder a meaningful analysis. First, given the vast extent of introgressions and the unclearness of their origin and timing, which extends beyond recent European history, there is too little information to build demographic models. Second, we have a strong sampling bias in the selection of the samples, which was designed to maximize diversity in the pan-genome. Thus our sampling does not reflect the present or past allele frequencies (haplotype length frequencies), which however would be necessary for demographic inference.

2. It was unclear from the supplementary methods exactly how gene models were lifted over between contigs. Was a single genome copy annotated and then these annotations were lifted over to the other haploid chromosomes or were all copies annotated and lift over was done amongst all copies to fill in annotation gaps? The former approach would seem to miss a fair number of genes present within a particular genome as PAV across homologous chromosomes.

In the updated version of the Supplementary methods, we now describe these points in more detail. In brief, we *de novo* predicted gene models for each haplotype individually. For efficiency reasons, we annotated the gene models on the contig-level assemblies (see "Initial tetraploid genome assembly and purging" in the Supplementary methods). Once the assemblies were finished, we used *liftoff* to transfer the gene models from the unphased contig assemblies to the respective final phased assemblies, using strict alignment parameters (alignment covers the complete length of the gene model without mismatches) to prevent mismapping of gene models to homologs that are present in different haplotypes.

3. The low-level of haplotype assignment (48%) in the test case for the pan-genome seems concerning given the intention for this to serve as resource for haplotype construction.

We agree that the prototype described in the original version of the manuscript only targeted specific regions of the genome. Even though the reconstruction of the targeted regions worked well, it did not target the entire genome. We have now drastically advanced our genome reconstruction method.

The method is now based on *all* haplotype-specific *k*-mers in *all* regions of the genome. With this we can reconstruct ~80% of the haplotypes of the genome of a modern potato cultivar using short reads only. Considering the repetitiveness of potato genomes and the size of the pericentromeric regions, we would like to emphasize that reconstructing 80% of the individual haplotypes of a tetraploid genomes with short reads is unprecedented.

We now demonstrate the usage of the graph in three different scenarios. In the first case, we reconstructed the genome of 'White Rose', a cultivar that was used to build the graph. In this optimistic use-case, we reconstructed haplotypes on a Mb scale implying that it is possible to reconstruct haplotype-resolved genome sequences with short reads if they are included in the graph. In the second case, we reconstruct the genome of 'Kenva', a cultivar that was derived from three cultivars that are included in the graph. We therefore expect that all haplotypes of 'Kenva' are in the graph, but that they are recombined. This simulated a realistic future use case, once we can use this method with a complete pan-genome. Again, the reconstruction of the Kenva genome led to Mb-scale haplotype-resolved regions.

The third use case is the most realistic use case. Here we used the genome-graph to reconstruct the genome sequence of 'Russet Burbank' with short reads only. 'Russet Burbank' is one of the most important potato cultivars and used for ~70% of the global French Fries production, but despite this importance its genome had not been assembled so far.

The reconstruction led to a genome assembly of 2.03 Gb with an N50 of 0.60 Mb. To validate the assembly, we have, in addition, generated a *de novo* genome assembly of 'Russet Burbank' (following the same procedure as we did for the 10 other cultivars, which led to a chromosome-level, haplotype-resolved assembly). We aligned the contigs from the genome-graph reconstruction to the *de novo* assembly and found that almost 90% of the contigs fully aligned to individual haplotypes of the *de novo* assembly, including an impressive, 9.9 Mb long contig (Supplementary Figure 31).

Taken together, the successful genome reconstruction of modern potato cultivars using the haplotype-graph outlined that the pan-genome will serve as a cost-effective method to analyze the genomes of large panels of potatoes.

Construction of crop pan-genomes is a very active area of research with several graph pan-genomes published over the last few years. A few examples include:

<https://www.sciencedirect.com/science/article/pii/S2590346223003498>

<https://genomebiology.biomedcentral.com/articles/10.1186/s13059-023-03133-2>

<https://www.nature.com/articles/s41422-022-00685-z>

Many of these studies are more comprehensive in terms of global diversity of their respective crops than the current study; the pan-genome described here will likely serve as an effective resource for European potato only.

We agree that there is a series of interesting studies based on pan-genome graphs across several crop species, which covered diversity analysis from sequence to gene level. We are now referencing this work in the manuscript, and have greatly revised our analysis of genetic diversity.

Having said this, the potato pan-genome will not only serve the reconstruction of European potatoes. Potatoes were first introduced to Europe and from there subsequently spread to many other parts of the world such as North America, Africa and Asia. Interestingly, one of the genomes that we reconstructed with the haplotype-graph, 'Russet Burbank', is in fact an US American cultivar. This proves that the genome graph can serve as an effective resource also for non-European potato genomes.

Furthermore, I do not see innovations in the approaches included here that push the field forward or that would be widely adopted across systems.

We respectfully disagree with your assessment of the innovation of our work. In fact, we introduce a novel haplotype-graph structure, which is an effective and practical solution to build up genome-graphs from many individual genomes. This graph can be used to reconstruct complex, tetraploid potato genomes (and any other inbred crop species). The contigs of such a pan-genome-based assembly can reach megabase-level even though they are based on short read sequencing only. We do not know any other method that would be able to do this in a polyploid.

In summary, I think this is a well-executed study, I believe the authors have generated resources that will be very useful for the European potato breeding community, and I

believe the manuscript is well-written and the results are clear. The study would find a good publication home in a more disciplinary journal.

Thank you again for your input and time to point out that the manuscript is well-written and clear. We still would like to state that this is the first study introducing a phased pan-genome of an autotetraploid species. We resolved and explained the co-occurrence of extreme sequence diversity (higher than in any crop species reported to date) and low haplotype diversity. We find patterns of selection and admixture. Ultimately, we use the resource to introduce a new idea for genome reconstruction based on haplotype-graphs, which is based on the fact that modern potatoes are derived from a narrow haplotype space including long, shared haplotype blocks. Taken together, we think that this manuscript presents multiple reasons why it should be published in Nature.

Referee #3 (Remarks on code availability):

Analyses and computational pipelines are well-documented in the accompanying repository.

Thank you very much for taking the time to check this.

Referee #1

I thank Sun et al. for their response to my comments, as well as those of the other reviewers. I believe that the incorporation of the reviewers' suggestions has improved the quality of the manuscript. The bioinformatics section of the manuscript is of high quality, and the new validations of the haplotype-graph method are truly impressive. However, the evolutionary genetics section of the manuscript is still unsatisfactory, and the authors' responses fall short of addressing the concerns or are not correctly interpreted. I will describe below the main points of my critique and my concrete suggestions to get this manuscript published:

Thank you very much for taking the time to read our manuscript again and for your helpful advice. Following your comments, we have added an update to the selection and introgression analyses.

Major comments:

-Positive selection

1) My original critique stated that reduction of diversity in conjunction with negative Tajima's D is not sufficient evidence to claim that a particular region of the genome has undergone positive selection and that a negative Tajima's D does not necessarily imply positive selection. To reply to my critique the authors investigated the relationship between genetic diversity and recombination rate, as well as genetic diversity and LD, observing a significant positive and negative R^2 , respectively (Fig. S29A-B). The positive correlation between genetic diversity and recombination rate has been recurrently observed among different taxa (reviewed in Kern and Hahn, MBE, 2018; but see also Corbett-Detig, PLoS Biology, 2015). This positive correlation is interpreted as the effect of linked selection on polymorphism. Linked selection can be the result of positive selection (hitchhiking) and/or purifying selection (background selection). Disentangling between these two alternatives across the genome is extremely difficult, but linked selection needs to be invoked to explain the patterns of Fig. S29A-B. Linked selection was not explicitly mentioned by the authors in their response or incorporated in the manuscript.

Thank you for making this clear. We agree to your point, the different types of linked selection (positive vs. background selection) cannot be distinguished in our analysis. We now explicitly state this in the manuscript and cite the relevant literature on genetic diversity and recombination rate.

2) In contrast to what the authors claim: "To test the effects of demography on genomic diversity, we correlated LD values with genetic diversity as well as with local

recombination rate along the chromosomes”, they have not tested the effects of demography at all. The above mentioned positive correlation between genetic diversity and recombination rate will be present with or without the presence of bottlenecks. Naturally, levels of LD are influenced by demography—population contractions increase LD, while expansions decrease it. Therefore, it stands to reason that repeated bottlenecks have raised LD levels, making it more difficult to detect positive selection. Formally testing the influence of demography would require explicitly modeling demographic scenarios and comparing them against your data. However, I am not suggesting that the authors take this approach.

We agree that the correlations themselves do not test for the effects of demography, but that they account for the combined effects of both demography and linked selection. We have now reworded this section.

The primary aim of this section was to identify regions in the genome that are under selection. We used the genome-wide correlations of genetic diversity, LD, and recombination rate to generate a baseline for the identification of outlier regions where the local values do not follow the global trends. Specifically, the outlier regions (identified with the residuals of a genome-wide mixed model analysis) highlight genomic regions that exhibit unusually high LD, suggesting particularly strong linked selection. Reassuringly, such regions were concentrated in the candidate regions with reduced genetic diversity, primarily in the region on chromosome 10.

3) Finally, the mixed model used by the authors to quantify the contributions of local recombination rate, the presence of structural variants, and chromosome identity to LD is not sufficient to support a claim of positive selection. The LD that is not explained by these factors could be in part caused by linked selection (positive and purifying selection in unknown proportions). The length of the putatively selected region on Chr10, along with its pericentromeric location, makes it unlikely that positive selection is the underlying cause of the observed pattern, since the selection coefficient that would be required to produce a sweep signal of this magnitude likely needs to be very high.

Similar as in our response to your first point, we agree that our analysis does not present direct evidence to claim positive selection. We widened our interpretation to *linked* selection in the manuscript. We would like to point out that this does not affect our main conclusions (that selection acted on the potato genomes during domestication and/or the transition to Europe).

4) In the rebuttal, the authors refer to purifying selection as the reason for the reduced level of genetic diversity and lack of introgression in the region of Chr10. This is the only instance, where the authors mention purifying selection, which is not mentioned

in the manuscript at all. It is confusing since in all other instances the authors seem to be referring to positive selection, when they say selection. As I mentioned above, I believe purifying selection is responsible for some unknown proportion of the pattern.

As mentioned, we fully agree that we cannot make a clear distinction between purifying and positive selection. We modified the manuscript to make clear that not only positive selection, but also purifying selection could be the underlying cause.

5) My suggestion is to remove this entire section from the manuscript. Rather than adding value to the excellent bioinformatics work, it undermines it. The absence of any reference to this section in either the abstract or the discussion further supports my suggestion to remove it from the manuscript.

We politely disagree. Discussing this work with the potato research community made it clear that this is an important aspect of this study. However, given your comments and the fact that we cannot identify the cause of the selection, we decided to make this section more concise.

-Introgression

6) I appreciate that the authors present a quantitative analysis of introgression using D-statistics and f₄-statistics. However, the analyses are not reported using current standards and the significance is not assessed in the proper way. D-statistics and f₄-statistics are meant to identify genome-wide patterns left by introgression. The significance of the test is evaluated by a block-jackknife procedure, which is meant to correct for the extent of LD in the assessed individuals/populations.

The authors should clearly present the tree configuration used in the D-statistics as (A,B;C,D), in which D is the outgroup, C is the candidate for introgression and A and B are the evaluated individual/populations. It is particularly important to report the outgroup used in the analysis, since the test requires the outgroup to be far enough and equally distant from the tested populations. The analysis is difficult to follow due to the non-standard way in which the analyses are reported.

You should only report genome-wide tests and specify the size of the jackknife blocks used in the analysis. The blocksize should be longer than the genome-wide LD decay. Window-based D-statistics should be avoided, as there is now way to test for statistical significance for each window, as the jackknife test of significance cannot be performed. None of the other D-statistic-inspired tests are adequate to be performed on windows.

Thank you for raising this important point. We agree that the window-based analysis was limiting the calculation of significance. Therefore, we have reanalyzed the data to evaluate the significance of the introgression tests at chromosome level. The new results confirmed that there were frequent introgressions into many of the haplotypes. We have added the details of these tests to the methods section including the tree configuration used for the D-statistics (in Fig 3). The results for each of the tests are shown in a new supplementary figure (Supplementary Figure 27).

As suggested, we also revised Fig. 3 to present the raw D-statistics estimates instead of categorizing regions as being introgressed or not. The display of D-statistics at window level is not a new approach and has been employed before (for example, Martin *et al.* 2019, PLOS Biology or Fu *et al.* 2022, Nature Ecology & Evolution). While we understand that individual regions cannot be tested for significance, they allow for comparative analyses of haplotypes, which revealed that large parts of the sequence differences between the haplotypes can be explained by introgressions (Fig. 2).

Finally, we also tested the effect of different block sizes (from 100 kb to 4 Mb). Even though the range of block sizes was large, it did not have a major effect on the outcome of the introgression tests. This is also described in the new method section.

7) The interpretation of the Admixture analysis presented in Figure 3b is not adequate. The authors stated that “genome-wide admixture analyses revealed clear signatures of admixture specific to the European cultivars.” Admixture/Structure plots cannot be used to infer a history of admixture/introgression, despite their suggestive names. The software only tries to minimize LD and Hardy-Weinberg disequilibrium for a given number of Ks and infer ancestry components. Multiple scenarios not involving introgression can give rise to patterns of mixed ancestry in populations, i.e. inclusion of diverse populations and subpopulations that underwent population bottlenecks.

We have revised the interpretation of the admixture analysis. In brief, we describe the observed patterns as reflecting mixed ancestry between European cultivars and the clade C4S (at $K \geq 3$). Along with this, we have removed any suggestion that the admixture plots alone provide evidence for introgression and ensured that their interpretation aligns with their methodological purpose.

8) Please indicate in the main text the optimal K that was identified.

The optimal number of ancestral populations (K) was now added to Fig. 3b.

9) In summary, the introgression section needs extensive reanalysis and needs to be rewritten.

We have updated the section and addressed all the points raised above.

Other comments:

Figure 3

Figure 3A: Clearly indicate in the phylogeny which species belong to clade C4N and C4S.

Done.

Figure 3D-E can be removed, as the per-window based tests are not recommended (as explained above). You can replace this by the equivalent but only for genome-wide tests as described above (reporting the tree configuration, significance and jackknife block size).

As per your suggestion, we have removed Fig. 3e, and added and modified Figs. 3d and 3e (according to our descriptions above).

Referee #2

This revised version has been substantially improved. I only have a few minor comments remaining.

Thank you very much for your supportive comments on our manuscript.

Haplotype sharing: The details on how shared haplotypes are defined seem to be missing. Are two haplotype blocks with identical sequences considered shared? Please clarify.

Two haplotype blocks (10kb) were considered identical if they did not include more than ten different sites (<0.1%). The threshold was selected based on the distribution of the number of segregating sites (NSS) between haplotype pairs (Fig. 2d and new Supplementary figure 48). The selected threshold allowed for subtle differences (e.g., assembly errors) between haplotypes, while it defined actually divergent haplotypes as distinct. We have added further details to the methods section.

The authors predicted between 36,862 and 47,316 protein-coding genes in the 40 haploid genomes. This large variation in the number of predicted genes should be explained or discussed. (The editor later forwarded your suggestion to lift over genes from some haplotypes to other haplotypes to identify missing (unannotated) genes and to refine the predicted gene models.)

The large variation in gene number between different potato genomes has been reported before. For example, similar differences have been found within the pan-genome of wild potato species (Tang *et al*, 2022, Nature). To understand the reason for these differences, we followed your suggestion and used liftoff to annotate putatively missing gene models between the individual haplotypes. For this, we mapped the gene models of the genome with the highest number of genes (Flava) to all other genomes. The large differences in gene number between the haplotypes remained (minimum: 38,703 genes, maximum: 46,312 genes, excluding Flava), implying they are not the result of annotation artefacts. We added this point to the Supplementary Information ("Gene annotation and BUSCO analysis").

In addition, we found that most of the differences could be explained by copy number variation clustered in certain genomic regions. Using short read alignments and read depth analysis, we found that these regions show sequence copy number variation between the different genomes, further corroborating that the underlying differences in gene number are real. A thorough description of the biological origin of this variation would certainly be interesting, but is outside the scope of our work.

Page 4: “multiple thousands contigs” -> “multiple thousands of contigs”.
“generate a recombinant population for each of the tetraploid genomes”: change
“genomes” to “cultivars”

Done.

Page 5, “Even though.....tuber flesh”: this is not a complete sentence.

Corrected.

Page 7: The term “chromosome identity” needs to be defined more clearly. What exactly does it refer to in this context?

Changed wording.

Page 23, legend of Fig. 3b: Please specify which chromosome’s result is shown here.

Thank you for pointing this out. We added this.

Referee #1

I thank Sun et al. for their response to my comments and for presenting a more concise manuscript.

I comment here on their response to my comments on the selection and introgression section. Quoted text refers to Sun et al. response to my comment or their to new version of the manuscript.

Thank you very much for taking the time to comment on our manuscript again. Following your suggestions, we have removed the selection section and updated the introgression analysis section.

-Selection

The selection section is more concise, with its interpretation more nuanced, acknowledging that their approach cannot disentangle the linked effects of positive and purifying selection. Their claim of positive selection is now appropriately framed as speculative: “Even though linked selection includes the combined effects of purifying and positive selection, we speculated that the latter played an important role in chromosome 10”.

I still believe this section adds very little to the manuscript, as detecting the footprints of linked selection is unsurprising. Moreover, their approach fails to distinguish between the linked effects of positive and purifying selection or to identify candidates of positive selection.

In their response the authors state that “We would like to point out that this does not affect our main conclusions (that selection acted on the potato genomes during domestication and/or the transition to Europe)”. This statement is very likely to be true—selection is constantly acting on genomes. I also agree that positive selection likely played a role during domestication and the transition to Europe. However, the challenge lies in detecting the footprints of positive selection and identifying the loci it targeted. The authors fell short of achieving these objectives.

As I mentioned in my previous comments, the authors still do not make any reference to the selection section in the abstract, further supporting the little relevance of this part for the overall study. I leave the decision regarding the inclusion of this section to the authors and the editor.

We have removed the selection section from the manuscript.

-Introgression

The reporting D-statistics and F4 statistics used in the revised manuscript now follows the standards of the field. The authors used two different configurations for the D-statistics:

D(C4S, C4S; Cultivar, C3) (they called this configuration C4S).

D(Cultivar, C4S; C3, C1+2) (they called this configuration C3).

Configuration I. follows the tree topology presented in figure 3a. This D-statistics uses C3 as an outgroup, thus the test assumes that C3 cannot take part in any introgression and always represents the ancestral state. In contrast, configuration II. uses C3 as a possible contributor to the introgression. Both three topologies I. and II. cannot be true at the same time. **In my opinion topology I. is more appropriate**, and if this is the case, topology II. violates the assumptions of the test and is an invalid use of the test.

I suggest removing configuration II. Moreover, a more standard way to organize the configuration would be D (Cultivar, Cultivar; C4S, C3), iterating along all possible samples of Cultivars and C4S and keeping C3 fixed.

The authors stated in their reply that “The display of D-statistics at window level is not a new approach and has been employed before”. **Indeed, it is not new but it is incorrect.** The expectation of $D=0$ if no introgression took place is valid only when averaged across many loci throughout the genome. As I explained in my last comments, it is not possible to test for the statistical significance of any identified region. D-statistics different from zero in genomic-windows can indicate local sorting of the discordant topologies used in the test and not necessarily introgression. Therefore, the test was developed as a genome-wide statistic, however the use of D-statistic at a chromosome scale is valid.

As per your suggestion, we removed the section on configuration II using C3 as a possible introgression source.

Regarding the suggested configuration D (Cultivar, Cultivar; C4S, C3), while more standard, it would not apply in regions where both cultivars contain C4S introgressions, making the topology unclear. Since all cultivar haplotypes show significant amounts of C4S introgressions in varying proportions, this configuration would test relative introgression levels rather than presence or absence. In contrast, our approach ensures that P1 and P2 (two C4S species) are consistently more related to each other than to P3 (which is the cultivar that is expected to be derived from C4N).

We acknowledge the concerns about window-level D-statistics. In the revised manuscript, we do not use them to infer introgression. Significance testing is based solely on chromosome-level analyses. At the window level, we present D-values to show their genomic distribution and variation across haplotypes, even when $D = 0$. The D-statistic, as a ratio of ABBA-BABA, is reported as is (like any other genome scan statistics), without claims of significance.

Referee #2

The authors have addressed all my concerns. I congratulate the authors for a very interesting study!

Thank you very much!